# Constitutive deficiency of the neurogenic hippocampal modulator AP2γ promotes anxiety-like behavior and cumulative memory deficits in mice from juvenile to adult periods

Eduardo Loureiro-Campos[1,2], António Mateus-Pinheiro[1,2], Patrícia Patrício[1,2], Carina Soares-Cunha[1,2], Joana Silva[1,2], Vanessa Morais Sardinha[1,2], Bárbara Mendes-Pinheiro[1,2], Tiago Silveira-Rosa[1,2], Ana Verónica Domingues[1,2], Ana João Rodrigues[1], João Oliveira[1,2,3], Nuno Sousa[1,2], Nuno Dinis Alves[1,2*†‡], Luísa Pinto[1,2*†]

[1]Life and Health Sciences Research Institute (ICVS), School of Medicine, University of Minho, Braga, Portugal; [2]ICVS/3B's -PT Government Associate Laboratory, Guimarães, Portugal; [3]IPCA-EST-2Ai, Polytechnic Institute of Cávado and Ave, Applied Artificial Intelligence Laboratory, Campus of IPCA, Barcelos, Portugal

*For correspondence:
nda2114@cumc.columbia.edu
(NDA);
luisapinto@med.uminho.pt (LP)

†These authors contributed equally to this work

Present address: ‡Department of Psychiatry, Columbia University, NY 10032, New York, United States

**Abstract:** The transcription factor activating protein two gamma (AP2γ) is an important regulator of neurogenesis both during embryonic development as well as in the postnatal brain, but its role for neurophysiology and behavior at distinct postnatal periods is still unclear. In this work, we explored the neurogenic, behavioral, and functional impact of a constitutive and heterozygous AP2γ deletion in mice from early postnatal development until adulthood. AP2γ deficiency promotes downregulation of hippocampal glutamatergic neurogenesis, altering the ontogeny of emotional and memory behaviors associated with hippocampus formation. The impairments induced by AP2γ constitutive deletion since early development leads to an anxious-like phenotype and memory impairments as early as the juvenile phase. These behavioral impairments either persist from the juvenile phase to adulthood or emerge in adult mice with deficits in behavioral flexibility and object location recognition. Collectively, we observed a progressive and cumulative impact of constitutive AP2γ deficiency on the hippocampal glutamatergic neurogenic process, as well as alterations on limbic-cortical connectivity, together with functional behavioral impairments. The results herein presented demonstrate the modulatory role exerted by the AP2γ transcription factor and the relevance of hippocampal neurogenesis in the development of emotional states and memory processes.

## Editor's evaluation

The aim of this study was to examine the impact of AP2γ deficiency on the development of sensorimotor skills, cognitive, and emotional function. This paper will be of interest to scientists in the fields of developmental psychobiology and neurogenesis.

## Introduction

New cells are continuously generated, differentiated into neurons, and integrated into the preexisting neural networks in restricted regions of the postnatal mouse brain (***Dennis et al., 2016***; ***Kempermann***

*et al., 2018*; *Moreno-Jiménez et al., 2019*; *Tobin et al., 2019*). One of these so-called neurogenic niches is the subgranular zone (SGZ) of the hippocampal dentate gyrus (DG). Here, neural stem cells (NSC) give rise to mature neural cells including glutamatergic granular neurons in a finely tuned process with many developmental steps sensitive to different regulatory influences (*Kempermann et al., 2004*; *Mateus-Pinheiro et al., 2017*; *Tobin et al., 2019*; *Toda et al., 2019*). The last years have seen an increase in the number of studies on the functional specificity of hippocampal neurogenesis during development (DHN) and adult hippocampal neurogenesis (AHN). The ontogenetic interpretation of hippocampal neurogenesis assigns different functional relevance to DHN as the neurogenic process that establishes the basic repertoire of adaptable behaviors and AHN as the neurogenic process that underpins the adult brain's ability to adapt functional behaviors (*Abrous et al., 2021*; *Masachs et al., 2021*; *Tronel et al., 2015*). This functional dissociation between these two stages of hippocampal neurogenesis allows us to understand how influences in the neurogenic process at different phases can lead to distinct impairments in emotional states and cognitive functions.

Postnatal hippocampal glutamatergic neurogenesis exhibits a regulatory transcriptional sequence (Sox2→ Pax6→ Ngn2→ AP2γ→ Tbr2→ NeuroD→ Tbr1) that recapitulates the hallmarks of the embryonic glutamatergic neurogenic process in the cerebral cortex (*Hochgerner et al., 2018*; *Mateus-Pinheiro et al., 2017*; *Nacher et al., 2005*). Transcriptional factors as Pax6, Ngn2, Tbr2, NeuroD, and Tbr1 have several and distinct roles in proliferation, cell kinetics, fate specification, and axonal growth (*Englund et al., 2005*; *Götz et al., 1998*; *Hevner, 2019*; *Hevner et al., 2006*; *Hochgerner et al., 2018*). Despite several efforts to understand the complex transcriptional network orchestration involved in the regulation of neurogenesis, both in early developmental stages and during adulthood, these are still to be fully understood (*Bertrand et al., 2002*; *Brill et al., 2009*; *Englund et al., 2005*; *Hack et al., 2005*; *Hsieh, 2012*; *Mateus-Pinheiro et al., 2017*; *Waclaw et al., 2006*). Recently, the transcription factor activating protein two gamma (AP2γ, also known as Tcfap2c or Tfap2c) was described to be an important regulator of glutamatergic neurogenesis in the adult hippocampus, being involved in the regulation of transient amplifying progenitors (TAPs) cells (*Hochgerner et al., 2018*; *Mateus-Pinheiro et al., 2018*; *Mateus-Pinheiro et al., 2017*). AP2γ belongs to the AP2 family of transcription factors that is highly involved in several systems and biological processes, such as cell proliferation, adhesion, developmental morphogenesis, tumor progression, and cell fate determination (*Eckert et al., 2005*; *Hilger-Eversheim et al., 2000*; *Thewes et al., 2010*). In addition, AP2γ is functionally relevant during embryonic neocortical development, playing detrimental roles in early mammalian extraembryonic development and organogenesis, being one of the molecular components regulating the number of upper layer neurons during ontogeny and phylogeny in the developing cortex (*Pinto et al., 2009*). AP2γ is critical for the specification of glutamatergic neocortical neurons and their progenitors, acting as a downstream target of Pax6 and being involved in the regulation of Tbr2 and NeuroD basal progenitors' determinants (*Mateus-Pinheiro et al., 2017*; *Pinto et al., 2009*). Strikingly, in humans, defects in the AP2γ gene were reported in patients with severe pre- and post-natal growth retardation (*Geneviève et al., 2005*), and to be involved in the mammary, ovarian and testicular carcinogenesis (*Hoei-Hansen et al., 2004*; *Li et al., 2002*; *Odegaard et al., 2006*).

AP2γ deletion during embryonic development results in a specific reduction of upper layer neurons in the occipital cerebral cortex, while its overexpression increases region- and time-specific generation of neurons from cortical layers II/III (*Pinto et al., 2009*). AP2γ expression persists in the adult hippocampus, particularly in a sub-population of TAPs acting as a positive regulator of the cell fate modulators, Tbr2 and NeuroD, and therefore as a promoter of proliferation and neuronal differentiation (*Mateus-Pinheiro et al., 2017*). Conditional and specific downregulation of AP2γ in the adult brain NSCs decreases the generation of new neurons in the hippocampal DG and disrupts the electrophysiological synchronization between the hippocampus and the medial prefrontal cortex (mPFC). Furthermore, mice with conditional deletion of AP2γ exhibit behavioral impairments, particularly a deficient performance in cognitive-related tasks (*Mateus-Pinheiro et al., 2018*; *Mateus-Pinheiro et al., 2017*). These studies reveal the crucial modulatory role of AP2γ during embryonic cerebral cortex development as well as its influence on glutamatergic neurogenesis and hippocampal-dependent behaviors during adulthood. Still, it is relevant to assess the importance of AP2γ for postnatal neurophysiology and behavior since development, comprehending how defects in the neurogenic process since early development can impact on behavior at specific developmental stages.

Herein, we explored the neurogenic, behavioral, and functional effects of AP2γ constitutive and heterozygous deletion in mice from early postnatal development to adulthood. We revealed that AP2γ deficiency promotes downregulation of the hippocampal glutamatergic neurogenic process, compromises functional limbic-cortical connectivity, and leads to significant behavioral impairments on emotional and memory functional modalities. We showed that these impairments induced by AP2γ deficiency do not occur only during adulthood and that compromised hippocampal neurogenesis leads to emotional and memory deficits as early as the juvenile phase. These functional behavioral impairments are either maintained from the juvenile phase to adulthood or emerge as behavioral flexibility and object location recognition deficits in adult mice with AP2γ constitutive deletion.

## Results

### Constitutive AP2γ deficiency decreases proliferation and neurogenesis in the postnatal DG, reduces dendritic length and complexity of short-DCX⁺ cells, without affecting neuronal morphology of mature granular neurons

We sought to dissect the impact of constitutive and heterozygous deficiency of AP2γ in the modulation of postnatal neuronal plasticity in the hippocampus, including its effect in the hippocampal neurogenic niche on proliferation and morphology of doublecortin⁺ (DCX) cells and pre-existing DG granule neurons (***Figure 1A***). In juvenile (P31) and adult mice (P70 – P84), we assessed the expression of markers for different cell populations (***Figure 1B***) along the neurogenic process through western blot and immunofluorescence. We observed a significant decrease in the expression levels of AP2γ protein in the hippocampal DG of juvenile (***Figure 1C***) and adult (***Figure 1D***) AP2γ heterozygous knockout mice (*Tfap2c⁺/⁻*, henceforth referred as AP2γ KO), with concomitant reduction of Pax6 and Tbr2 protein levels, but not Sox2, an upstream regulator of AP2γ (***Mateus-Pinheiro et al., 2017***).

Analysis of cell populations in the hippocampal neurogenic niche using BrdU and DCX labeling (***Figure 1E***) revealed that the number of BrdU⁺ and BrdU⁺/DCX⁺ cells is reduced in both juvenile and adult AP2γ KO mice, suggesting a decrease in the number of fast proliferating cells (TAPs) and neuroblasts, respectively (***Figure 1F–I***). Of note, we observed a decrease in these cell populations with age in both WT and AP2γ KO mice in agreement with previous reports (***Kase et al., 2020***; ***Katsimpardi and Lledo, 2018***; ***Supplementary file 1***).

To clarify the impact of AP2γ on the dendritic development and structural neuronal formation in the dorsal DG, we analyzed the dendritic structure and complexity of immature neurons based on DCX staining (***Figure 2***). DCX is a cytoskeletal protein expressed both in neuronal progenitors and immature neurons (***Dioli et al., 2019***; ***Francis et al., 1999***; ***Horesh et al., 1999***). We assessed the morphology of DCX⁺ cells only in adult animals as during juvenile period, the large amount of DCX⁺ cells does not allow a proper segmentation of dendrites through this methodology (***Figure 2A***). We divided the analysis of DCX⁺ immature neurons into those that exhibit dendrites that branch into the inner molecular layer (IML) – short-DCX⁺ cells – and DCX⁺ cells with dendritic trees that reach the medial/outer molecular layer (M/OML) – long-DCX⁺ cells, as previously described (***Dioli et al., 2019***; ***Figure 2B***). Adult AP2γ KO mice presented decreased density of both short- and long-DCX⁺ cells in comparison to WT animals (***Figure 2C and G***). Moreover, deficiency of AP2γ led to a reduced dendritic length (***Figure 2D and E***) and neuronal arborization (***Figure 2F***) of short-DCX⁺ cells. Although the density of long-DCX⁺ cells is reduced in AP2γ KO mice, dendritic length and arborization complexity of the existing long cells are similar to the WT animals (***Figure 2H-J***). Interestingly, when we analyzed pre-existing granular hippocampal neurons, constitutive deficiency of AP2γ did not alter their dendritic length and the neuronal arborization complexity either during juvenile period (***Figure 2—figure supplement 1A*** and C;) or at adulthood (***Figure 2—figure supplement 1B*** and C), when compared to WT mice. These observations suggest that AP2γ deficiency in adult mice delays the maturation of granular neurons but has no impact on their definitive morphology.

Taken together, the results herein revealed that AP2γ modulatory actions in the hippocampal neurogenic niche are similar and maintained in juvenile and adult mice. AP2γ transcription factor regulates NSCs proliferation and neuronal differentiation in the postnatal hippocampus by interacting with the different modulators involved in the transcriptional regulation of postnatal hippocampal

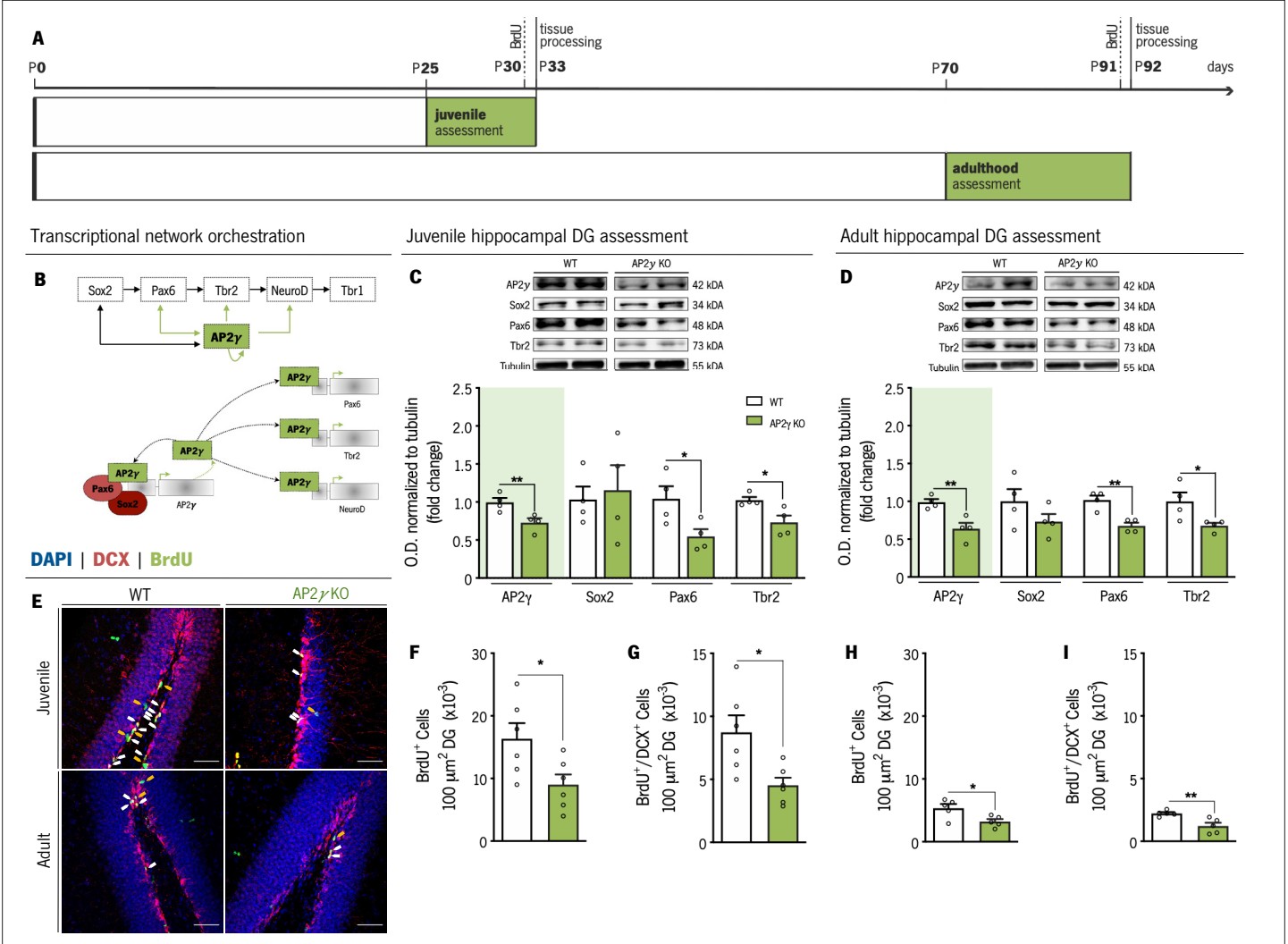

**Figure 1.** AP2γ constitutive and heterozygous deficiency reduces postnatal hippocampal neurogenesis both at juvenile and adult periods.
(**A**) Experimental timeline. (**B**) Transcriptional network of hippocampal neurogenesis under modulatory role of AP2γ. Western-blot analysis of AP2γ, Sox2, Pax6, and Tbr2 in juvenile (**C**) and adult (**D**) dentate gyrus (DG). (**E**) Hippocampal DG coronal sections stained for bromodeoxyuridine (BrdU) (green), doublecortin (DCX) (red), and DAPI (blue). BrdU+/DCX+ cells are indicated by white arrows and solely BrdU+ cell is identified with yellow arrows. (**F–I**) Cell counts of BrdU+ and BrdU+/DCX+ cells in the hippocampal DG of juvenile and adult mice. Data presented as mean ± SEM. Sample size: Western-blot analysis: $n_{WT\ juvenile}$ = 4; $n_{AP2\gamma\ KO\ juvenile}$ = 4; $n_{WT\ adult}$ = 4; $n_{AP2\gamma\ KOadult}$ = 4; Immunostainings: $n_{WT\ juvenile}$ = 6; $n_{AP2\gamma\ KO\ juvenile}$ = 6; $n_{WT\ adult}$ = 5; $n_{AP2\gamma\ KOadult}$ = 5 [Student's t-test; ** $p < 0.01$; * $p < 0.05$; Statistical summary in ***Supplementary file 1***]. Scale bars represent 50 µm. Abbreviations: WT, wild-type; AP2γ KO, AP2γ heterozygous knockout mice; O.D., optical density.

The online version of this article includes the following source data for figure 1:

**Source data 1.** Western-blot analysis at the juvenile and adult timepoints.

neurogenesis. Furthermore, it suggests an important role of AP2γ transcription factor for a timely maturation of dendritic branching in newly born granular neurons in adulthood.

## AP2γ KO mice display normal early postnatal development but anxiety-like behavior and memory impairments at juvenile period

In light of the negative impact of AP2γ heterozygous deficiency in the postnatal neurogenic process in the hippocampus, known as an important modulator of emotional and memory functions (***Christian et al., 2014***; ***Toda et al., 2019***), we assessed early postnatal development and the behavioral performance of WT and AP2γ KO at juvenile and adult periods.

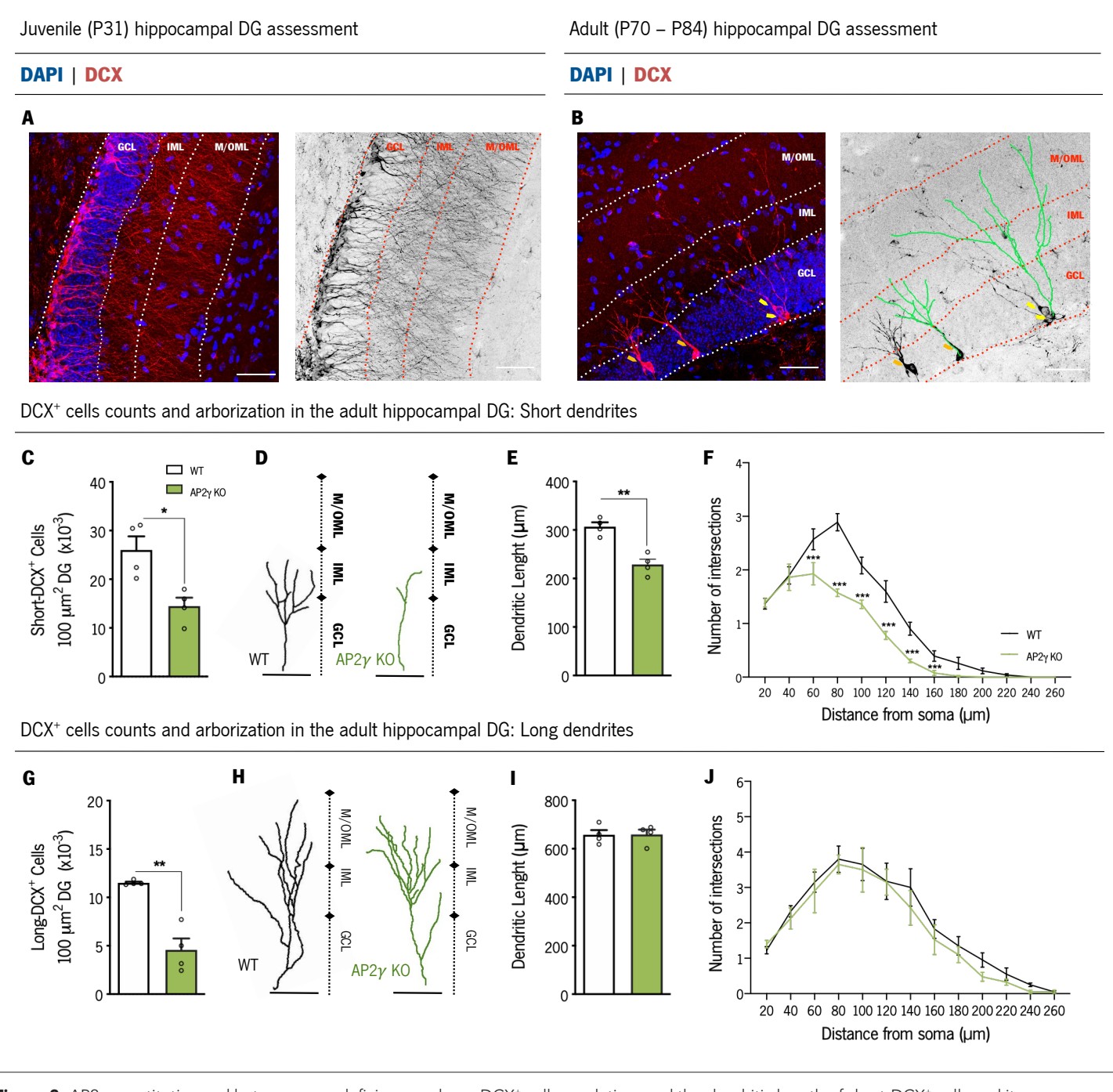

**Figure 2.** AP2γ constitutive and heterozygous deficiency reduces DCX⁺ cell population, and the dendritic length of short-DCX⁺ cells and its arborization complexity in adult mice. (**A and B**) Dorsal hippocampal DG coronal sections stained for doublecortin (DCX) (red) and DAPI (blue) with the corresponding color conversion images. These representative images include the DG subregions, the granular cell layer (GCL), the inner and medial/outer molecular layers (IML and M/OML, respectively). Short DCX⁺ cells are indicated by orange arrows and long DCX⁺ cells by yellow arrows. (**B**) Dendritic tree of short (left) and long (right) DCX⁺ cells are traced in green. Cell counts of short (**C**) and long (**G**) DCX⁺ cells. Dendritic length of reconstructed short (**D and E**) and long DCX⁺ (**H and I**) cells, and respective sholl analysis (**F and J**). Data presented as mean ± SEM. Sample size: $n_{WT\ adult}$ = 4; $n_{AP2\gamma\ KO adult}$ = 4. [Student's t-test and Repeated measures ANOVA; ***$p < 0.001$, ** $p < 0.01$, * $p < 0.05$; Statistical summary in **Supplementary file 1**]. Scale bars represent 50 μm. Abbreviations: WT, wild-type; AP2γ KO, AP2γ heterozygous knockout mice; P, Postnatal day.

The online version of this article includes the following figure supplement(s) for figure 2:

**Figure supplement 1.** AP2γ constitutive and heterozygous deficiency does not impact on the morphology of hippocampal granular neurons, both at the juvenile and adult periods.

**Table 1.** Results from the milestones protocol tests included in the assessment of early postnatal neurodevelopment.

Sample size: $n_{WT}$ = 9; $n_{AP2\gamma\ KO}$ = 9. Abbreviations: WT, wild-type; AP2 $\gamma$ KO, AP2 $\gamma$ heterozygous mice; P, Postnatal day.

| Milestone test | | WT Day (median) | AP2 $\gamma$ KO Day (median) | Statistical test, significance **Mann-Whitney test** | Typical range |
|---|---|---|---|---|---|
| Rooting | | 7 | 7 | $U$ = 31.50, p = 0.43 | P1 – P15 |
| Ear twitch | | 8 | 9 | $U$ = 24.50, p = 0.16 | P6 – P14 |
| Auditory startle | | 8 | 7 | $U$ = 37.50, p = 0.85 | P7 – P16 |
| Open field | | 8 | 8 | $U$ = 39.50, p > 0.99 | P6 – P15 |
| Walking | | 10 | 9 | $U$ = 38.00, p = 0.78 | P7 – P14 |
| Surface righting | | 7 | 7 | $U$ = 38.50, p = 0.88 | P1 – P10 |
| Negative geotaxis | | 8 | 9 | $U$ = 25.00, p = 0.17 | P3 – P15 |
| Cliff aversion | | 7 | 6 | $U$ = 26.00; p = 0.22 | P1 – P14 |
| Postural reflex | | 9 | 8 | $U$ = 26.00, p = 0.21 | P5 – P21 |
| Air righting | | 12 | 11 | $U$ = 29.00, p = 0.32 | P7 – P16 |
| Wire suspension | | 13 | 11 | $U$ = 21.50, p = 0.10 | P5 – P21 |
| Grasping | | 16 | 16 | $U$ = 30.00, p = 0.37 | P13 - P17 |
| Eye-opening | | 12 | 11 | $U$ = 9.00, p < 0.01 | P7 - P17 |
| | | WT Mean ±SEM | AP2γ KO Mean ± SEM | Statistical test, significance Repeated measures ANOVA | |
| Homing | Trial 1 | 6.89 s ± 0.44 | 6.89 s ± 1.32 | $F_{(1,48)}$ = 0.06, p = 0.80 | |
| | Trial 2 | 6.33 s ± 1.12 | 6.89 s ± 1.12 | | |
| | Trial 3 | 5.22 s ± 1.57 | 5.78 s ± 1.57 | | |
| Weight gain pattern | | 8.99 g ± 0.53 | 9.22 g ± 0.16 | $F_{(1,16)}$ = 0.32, p = 0.58 | |

Evaluation of early postnatal development was performed through the assessment of somatic and neurobiological parameters during the first 21 postnatal days (*Table 1*; *Figure 3—figure supplement 1*). Despite a variation in the eye-opening day, responsiveness in sensory-motor functions, vestibular area-dependent tasks, and strength, as well as somatic parameters were similar in WT and AP2γ KO mice. Furthermore, all analyzed parameters were within the previously described range (*Guerra-Gomes et al., 2021*; *Heyser, 2004*). These observations suggest that constitutive heterozygous deletion of AP2γ has no impact on early postnatal development.

We then performed an assessment of different behavioral domains in juvenile mice (P25 - P33; *Figure 3A*). The open-field (OF) and the novelty suppressed-feeding (NSF) tests were used to assess anxiety-like behavior. In the OF, juvenile AP2γ KO mice exhibited decreased distance traveled in the anxiogenic center of the arena, in comparison to WT animals, (*Figure 3B*) suggesting an anxiety-like phenotype promoted by AP2γ deficiency. Concomitantly, AP2γ KO mice exhibited an increased latency to reach and feed on the food pellet in the NSF test (*Figure 3C and D*). Of note, WT and AP2γ KO have similar locomotor activity (*Figure 3—figure supplement 2A*) and appetite drive (*Figure 3—figure supplement 2B*).

The assessment of adaptive behavior to cope with an inescapable stressor and self-care were then evaluated throughout the tail suspension test (TST) and the sucrose splash test (SST), respectively. We observed no impact of AP2γ constitutive and heterozygous deletion in these emotional domains, as no alterations were detected in the immobility time in the TST (*Figure 3E*) and grooming time in the SST (*Figure 3F*).

Furthermore, we assessed memory in juvenile mice through the Morris water maze (MWM), the object recognition test (ORT), the object-in-context recognition test (OIC), and the contextual fear conditioning (CFC) test. In the MWM, WT and AP2γ KO juvenile mice present a high and steady

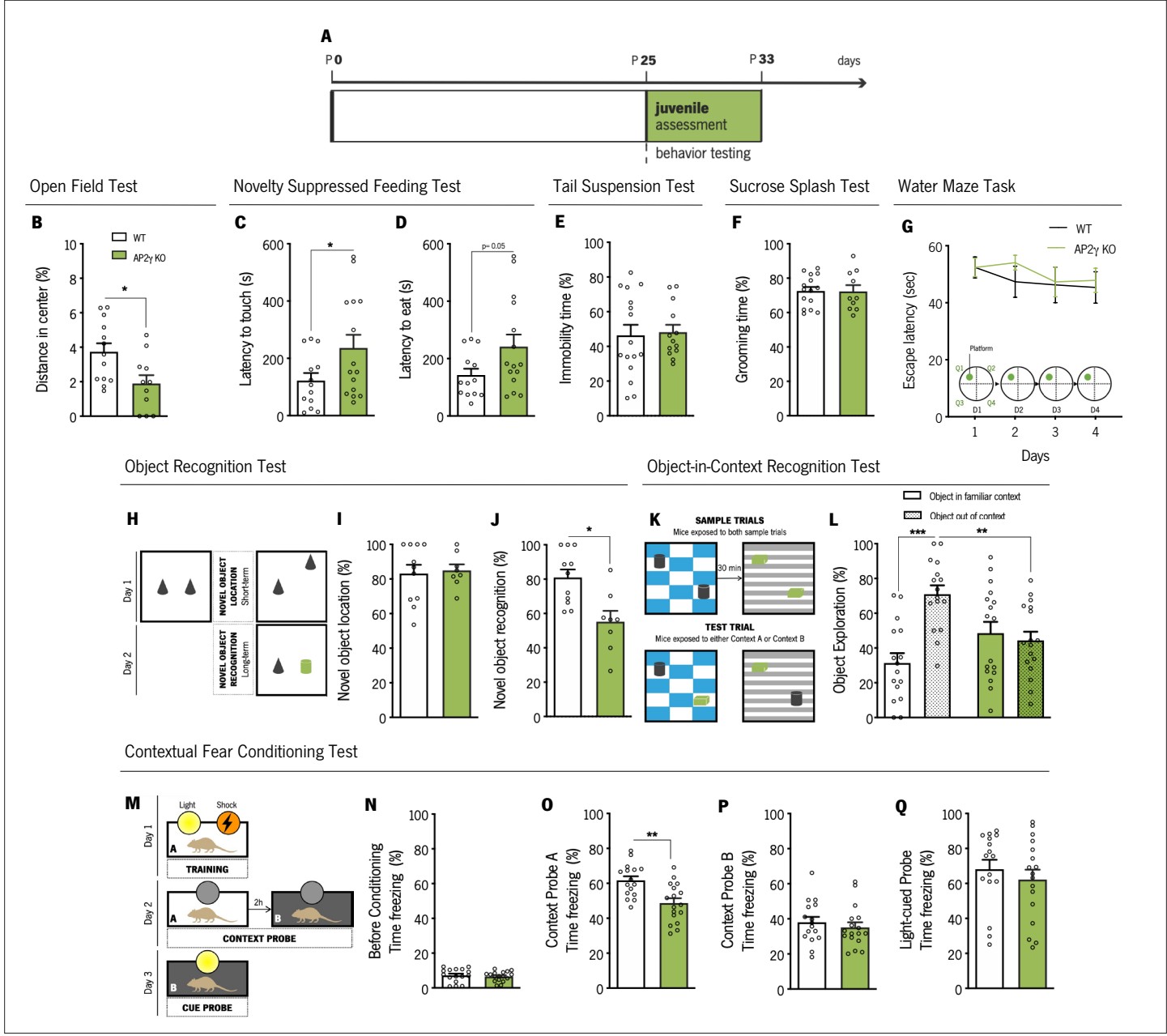

**Figure 3.** AP2γ constitutive and heterozygous deficiency increases anxiety-like behavior and promotes cognitive deficits in juvenile mice. (**A**) Timeline of behavioral assessment. Anxiety-like behavior was assessed through the open-field (OF) (**B**) and the novelty suppressed feeding (NSF) (**C and D**) tests. Coping and anhedonic-like behavior by the tail-suspension (TST) (**C**) and the sucrose splash (SST) (**D**) tests. (**E**) To evaluate cognitive behavior, juvenile mice were subjected to the Morris water maze (MWM) (**G**), the object recognition (ORT) (**H–J**), the object-in-context recognition (OIC) (**K and L**) and the contextual fear conditioning (CFC) (**M–Q**). Data presented as mean ± SEM. Sample size: OF: $n_{WT} = 13$; $n_{AP2γ KO} = 11$; NSF: $n_{WT} = 13$; $n_{AP2γ KO} = 15$; TST: $n_{WT} = 16$; $n_{AP2γ KO} = 13$; SST: $n_{WT} = 15$; $n_{AP2γ KO} = 10$; MWM: $n_{WT} = 10$; $n_{AP2γ KO} = 9$; ORT: $n_{WT} = 11$; $n_{AP2γ KO} = 8$; OIC: $n_{WT} = 16$; $n_{AP2γ KO} = 17$; CFC: $n_{WT} = 16$; $n_{AP2γ KO} = 17$. [Student's t-test, Repeated measures ANOVA and Two-way ANOVA; *** $p < 0.001$; ** $p < 0.01$; * $p < 0.05$; Statistical summary in ***Supplementary file 1***]. Abbreviations: WT, wild-type; AP2γ KO, AP2γ heterozygous knockout mice.

The online version of this article includes the following figure supplement(s) for figure 3:

**Figure supplement 1.** AP2γ constitutive and heterozygous deficiency does not impact on sensory-motor development.

**Figure supplement 2.** In juvenile mice, AP2γ constitutive and heterozygous deficiency does not impact on motor activity in the OF, nor on food consumption in the NSF test.

escape latency to find the hidden platform throughout training, implying that, at this age, mice are unable to learn this spatial navigation task at this early age (*Figure 3G*). This phenotype is consistent with the postnatal maturation state of the DG (*Hochgerner et al., 2018*; *Kozareva et al., 2019*).

In the ORT (Figure 3H), despite no differences in the novel object location (*Figure 3*I), AP2γ KO mice displayed significant deficits in the novel object recognition, as denoted by a decreased preference to explore the novel object (*Figure 3J*). In the OIC behavioral paradigm (*Figure 3K*), WT mice spent a greater proportion of time exploring the out of context object, than the object in the familiar environment, whereas AP2γ KO mice showed no preference of object exploration (*Figure 3L*). Thus, AP2γ KO mice were not able to successfully discriminate between the two distinct contexts, demonstrating an impaired PFC-hippocampus-perirhinal cortex network (*Barker and Warburton, 2020*).

In the CFC, a behavior test described to be sensitive to changes in hippocampal neurogenesis (*Gu et al., 2012*), juvenile mice were subjected to two distinct context tests, aimed to test hippocampal-dependent memory, and a cue probe to assess the integrity of extrahippocampal memory circuits (*Figure 3M*; *Gu et al., 2012*; *Mateus-Pinheiro et al., 2017*). As expected, prior to conditioning, freezing behavior was low, and WT and AP2γ KO mice identically explored the acrylic cylinder (*Figure 3N*). In context A, juvenile AP2γ KO mice exhibited reduced freezing behavior when exposed to a familiar context (*Figure 3O*). No alterations in freezing behavior were observed neither in context

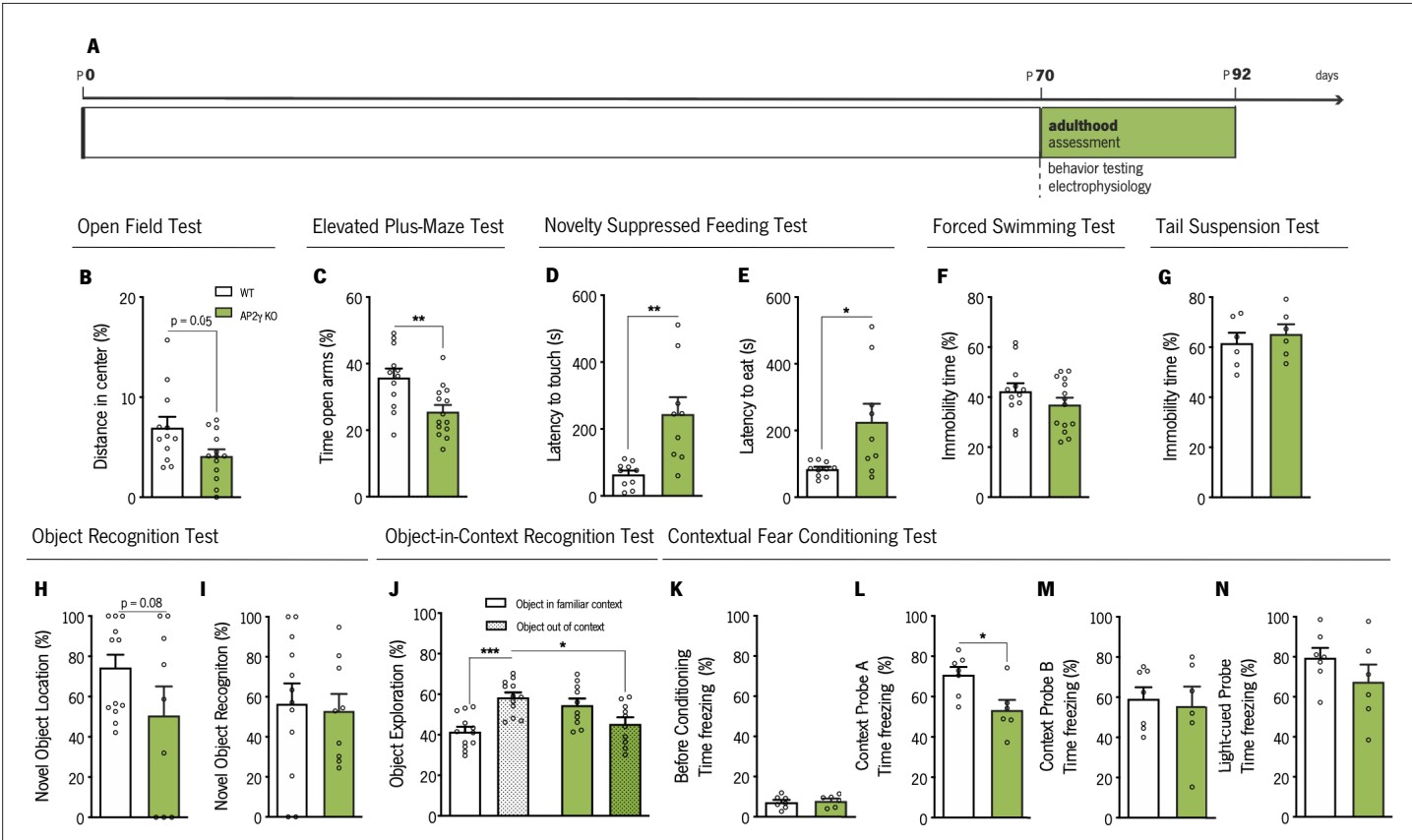

**Figure 4.** Behavioral assessment of adult mice. (**A**) Experimental timeline. Anxiety-like behavior was assessed through open-field (OF) (**B**), elevated plus-maze (EPM) (**C**) and novelty suppressed feeding (NSF) (**D and E**) tests, while coping behavior was evaluated through forced-swimming test (FST) (**G**) and the tail-suspension test (TST) (**G**). Object recognition test (ORT) (**H and I**), object-in-context recognition (OIC) (**J**) and the contextual fear conditioning (CFC) (**K–N**) were performed to assess cognitive behavior. Data presented as mean ± SEM. Sample size: OF, EPM and FST: $n_{WT}$ = 12; $n_{AP2γ KO}$ = 14; NSF: $n_{WT}$ = 10; $n_{AP2γ KO}$ = 9; TST: $n_{WT}$ = 6; $n_{AP2γ KO}$ = 6; ORT: $n_{WT}$ = 12; $n_{AP2γ KO}$ = 9; OIC: $n_{WT}$ = 12; $n_{AP2γ KO}$ = 10; CFC: $n_{WT}$ = 7; $n_{AP2γ KO}$ = 6. [Student's t-test and Two-way ANOVA; *** $p < 0.001$; ** $p < 0.01$; * $p < 0.05$; Statistical summary in *Supplementary file 1*]. Abbreviations: WT, wild-type; AP2γ KO, AP2γ heterozygous knockout mice.

The online version of this article includes the following figure supplement(s) for figure 4:

**Figure supplement 1.** In adult mice, AP2γ constitutive and heterozygous deficiency does not impact on motor activity in the OF, nor on food consumption in NSF test.

B (*Figure 3P*) nor in the cue probe (*Figure 3Q*). These CFC results, along with the OIC findings, reveal that AP2γ KO juvenile mice exhibit deficits in contextual memory.

Despite no evident impact on early postnatal development, AP2γ constitutive heterozygous deficiency impairments in hippocampal neurogenesis since development leads to a hypersensitivity toward aversive stimulus, as revealed by the presence of an anxious-like phenotype and memory impairments. These results highlight the involvement of the AP2γ transcription factor in the proper acquisition of important behavioral functions.

## Adult AP2γ KO mice maintain emotional and memory dysfunctions, while behavioral flexibility and object location recognition impairments emerge during adulthood

Adult WT and AP2γ KO mice were also tested for emotional and memory paradigms to determine if behavioral changes caused by the effects of AP2γ constitutive deletion on postnatal hippocampal neurogenesis were still present in adulthood (*Figure 4A*).

The OF, the elevated plus-maze (EPM), and the NSF tests were performed to evaluate anxiety-like behavior. In the OF test, AP2γ KO mice showed a trend toward a decrease in the distance traveled in the anxiogenic center of the arena ($P = 0.05$, *Figure 4B*), with no changes in locomotor activity as denoted by similar average velocity (*Figure 4—figure supplement 1A*). In the EPM test, AP2γ KO mice spent significantly less time in the open arms than WT mice (*Figure 4C*). These results are in agreement with results from the NSF test, where AP2γ KO mice displayed increased latency to reach and feed from the food pellet (*Figure 4D and E*). No changes in appetite drive were observed between groups, since they fed similarly in the post-NSF food assessment (*Figure 4—figure supplement 1B*). The forced swimming test (FST) and TST were then performed to examine the adaptive behavior to cope with inescapable stressors. WT and AP2γ KO mice exhibit similar immobility time in FST (*Figure 4F*) and TST (*Figure 4G*). These observations suggest that anxiety-like behavior promoted by constitutive AP2γ deficiency persist in adult mice, with no alteration in coping behavior.

Additionally, memory performance was evaluated by the ORT, OIC, CFC, and MWM. ORT revealed a trend toward a decrease in the preference to explore the displaced object of AP2γ KO when compared to WT mice ($p = 0.08$, *Figure 4H*). However, no alterations were observed in preference toward the novel object (*Figure 4I*). In the OIC paradigm, results were similar to juvenile mice. AP2γ KO adult mice were not able to discriminate between the two contexts, since they showed no preference of object exploration, whereas WT animals spent a greater proportion of time exploring the out of context object (*Figure 4J*). Prior to conditioning, freezing behavior in the CFC was low, and WT and AP2γ KO performed similarly (*Figure 4*K). AP2γ KO mice exhibited reduced freezing behavior when exposed to a familiar conditioning context - context A (*Figure 4L*), whereas no alterations in freezing behavior were observed neither in context B (*Figure 4M*) nor in the cue probe (*Figure 4N*). These CFC observations, along with the results from the OIC paradigm, suggest that adult AP2γ KO mice have deficits in contextual memory and an intact associative non-hippocampal-dependent memory.

The experimental groups were also subjected to the MWM test for evaluation of spatial memory (*Figure 5*). In the reference memory task, that relies on hippocampal function integrity (*Cerqueira et al., 2007*), AP2γ KO and WT mice exhibit similar performance to reach the hidden platform along the training days (*Figure 5A and B*). When the platform was changed to the opposite quadrant to assess behavior flexibility, a task that relies not only in hippocampal neurogenesis (*Anacker and Hen, 2017*; *Dupret et al., 2008*) but also in prefrontal cortical areas (*Hamilton and Brigman, 2015*), adult AP2γ KO mice spent less time in the new quadrant than WT animals (*Figure 5C*) suggesting that constitutive AP2γ deficiency leads to impaired behavioral flexibility. Detailed analysis of the strategies adopted to reach the escape platform (*Antunes et al., 2021*; *Garthe et al., 2009*; *Garthe and Kempermann, 2013*; *Mateus-Pinheiro et al., 2017*; *Ruediger et al., 2012*) revealed that AP2γ KO mice delayed the switch from non-hippocampal dependent ('Block 1') to hippocampal-dependent ('Block 2') strategies (*Figure 5D–H*), suggesting an impairment of hippocampal function. No differences were found in the working memory task (*Figure 5—figure supplement 1A* and B).

Overall, these results show that AP2γ constitutive deficiency negatively modulates not only early postnatal hippocampal neurogenesis but also AHN. These neurogenic dysfunctions result in emotional and memory deficits that persist in the adult mice, together with impairments in behavioral flexibility and object location recognition impairments that emerge during adulthood. Specifically, adult AP2γ KO mice

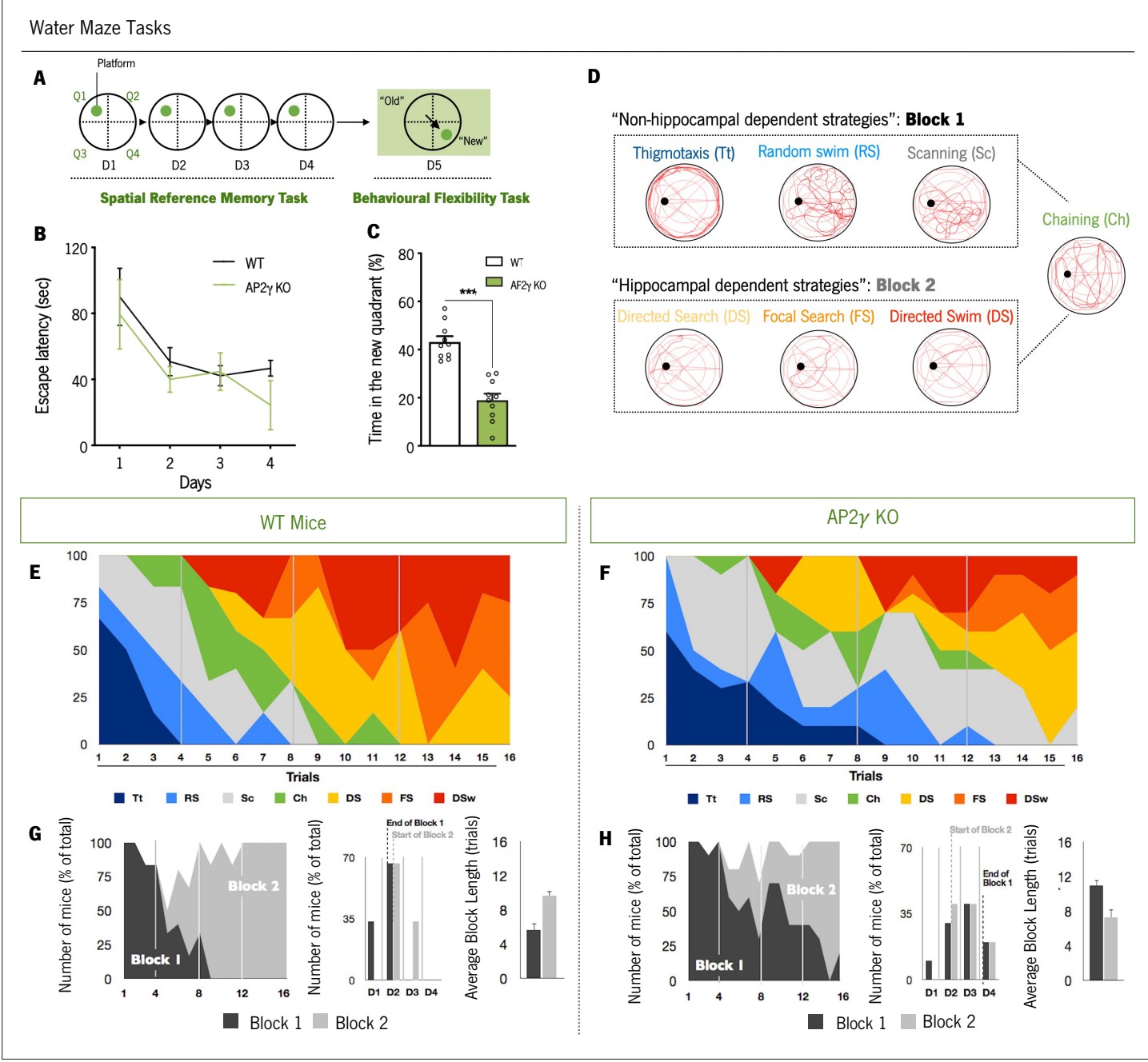

**Figure 5.** Cognitive performance of adult mice in the Morris water maze test. (**A and B**) Spatial reference memory was assessed as the average escape latency to find a hidden and fixed platform in each test day. (**C**) In the last testing day, animals were subjected to a reversal-learning task to test behavioral flexibility. (**D**) Schematic representation of typical strategies to find the platform during spatial memory evaluation grouped according to its dependence of the hippocampus (Block 1: Non-hippocampal dependent strategies; Block 2: Hippocampal dependent strategies). Average of each strategy used for WT (**E**) and AP2γ KO (**F**) mice, by trial number. The prevalence of each block along with trials, the distribution of strategies-block boundaries, and overall block length are shown for (**G**) WT and (**H**) AP2γ KO mice. Data presented as mean ± SEM. $n_{WT}$ = 10; $n_{AP2γ\ KO}$ = 10. [Repeated measures ANOVA and Student's t-test; ***$p < 0.001$; Statistical summary in *Supplementary file 1*]. Abbreviations: WT, wild-type; AP2γ KO, AP2γ heterozygous knockout mice.

The online version of this article includes the following figure supplement(s) for figure 5:

**Figure supplement 1.** In adult mice, AP2γ constitutive and heterozygous deficiency does not impact the working memory acquisition.

exhibit anxiety-like behavior, contextual-memory impairments, and deficiencies in behavioral flexibility capacities. Moreover, due to the relevance of the hippocampus and mPFC to memory-related behavioral

tests here assessed (OIC and MWM behavioral flexibility task) in adult mice, the functional integrity of these brain areas is suggested to be affected by constitutive and heterozygous deficiency of AP2γ.

## Adult hippocampal-to-PFC functional connectivity is disrupted by AP2γ constitutive and heterozygous deficiency

Given the impact of AP2γ deficiency on emotional behavior and memory, we sought for a functional correlate by investigating related neurocircuits. In adult WT and AP2γ KO mice, we explored the integrity of the hippocampus-to-medial prefrontal cortex (mPFC) circuitry, assessing electrophysiological features of local field potentials (LFPs) simultaneously in these connected brain areas (*Figure 6A* and *Figure 6— figure supplement 1A*). In AP2γ KO mice, the temporal structure of LFPs recorded simultaneously from the dorsal hippocampus (dHip) and mPFC was affected. Specifically the spectral coherence between these regions (*Adhikari et al., 2010*; *Oliveira et al., 2013*; *Sardinha et al., 2017*) in AP2γ KO mice is significantly decreased in delta, theta and beta frequency bands when compared to WT littermates (*Figure 6B*), suggesting the importance of AP2γ for an intact dHip-mPFC functional connectivity. AP2γ constitutive deficiency had a subtle impact in PSD values in the dHip, specifically in the Theta and Beta frequency bands (*Figure 6C*). In the mPFC, PSD values in delta, theta and beta frequencies bands were significantly lower in AP2γ KO than in WT mice (*Figure 6D*). In contrast, deficiency of AP2γ did not exert an effect neither in the spectral coherence between the vHip and the mPFC (*Figure 6—figure supplement 1B*) nor in the PSD values in the vHip (*Figure 6—figure supplement 1C*).

The electrophysiological studies revealed that AP2γ constitutive and heterozygous deficiency led to two outcomes: first, a significant decrease of coherence between the dHip and the mPFC indicating impairments in the ability of these regions to functionally interact; second, this decrease in interregional coherence was accompanied by a diminished neuronal activity in a range of frequencies in the mPFC, including in theta and beta frequencies, previously shown to be critically related with behavioral outputs dependent on cortico-limbic networks (*Colgin, 2011*; *Fell and Axmacher, 2011*; *Oliveira et al., 2013*).

## Discussion

Herein, we show that despite a normal early postnatal acquisition of neurodevelopmental milestones, AP2γ constitutive and heterozygous deficiency causes emotional and memory deficits in juvenile mice that persist into adulthood, while behavioral flexibility and object location recognition impairments emerge only in adult mice. These findings suggest that AP2γ is required for the proper development and maturation of neural circuits involved in relevant behavioral functions from early development to adulthood.

Newly generated neurons are highly relevant to hippocampal functioning and hippocampal-associated behaviors (*Anacker and Hen, 2017*; *Christian et al., 2014*; *Fang et al., 2018*; *Gonçalves et al., 2016*). Impairments in hippocampal neurogenesis precipitate the emergence of depressive- and anxiety-like behaviors (*Bessa et al., 2009*; *Hill et al., 2015*; *Mateus-Pinheiro et al., 2013a*; *Mateus-Pinheiro et al., 2013b*; *Revest et al., 2009*; *Sahay and Hen, 2007*). Here, we assessed for the first time the longitudinal impact of AP2γ, a transcription factor that plays an important role on embryonic neuronal development (*Pinto et al., 2009*) and recently described as a novel regulator of adult hippocampal neurogenesis (*Mateus-Pinheiro et al., 2018*; *Mateus-Pinheiro et al., 2017*), on neural plasticity, function and behavior at different postnatal periods.

Characterization of the neurogenic process in the hippocampal DG in juvenile and adult constitutive AP2γ KO mice, revealed that in agreement with our previous reports (*Mateus-Pinheiro et al., 2018*; *Mateus-Pinheiro et al., 2017*), AP2γ regulates upstream neurogenic regulators as Pax6 and Tbr2. Other modulators of the TAP's population, such as Ngn2 and Tbr2, have been shown to exert a similar control of hippocampal neurogenesis (*Galichet et al., 2008*; *Hodge et al., 2012*; *Roybon et al., 2009*; *Tsai et al., 2015*). Also, we observed, at juvenile period and during adulthood, that AP2γ plays an essential role in the regulation of pivotal neurogenic steps of NSCs proliferation and neuronal maturation. In addition to a decreased proliferation and neuroblasts population, AP2γ constitutive and heterozygous deficiency led to a reduced number of immature neurons and delayed morphological maturation of granular neurons in the adult dorsal hippocampal DG. Yet, these reduced neurogenesis and delayed maturation did not alter the definitive morphology of granular neurons, another form of hippocampal structural plasticity (*Bessa et al., 2009*; *Mateus-Pinheiro et al., 2013a*). Due to a large amount and density of DCX⁺ cells in the

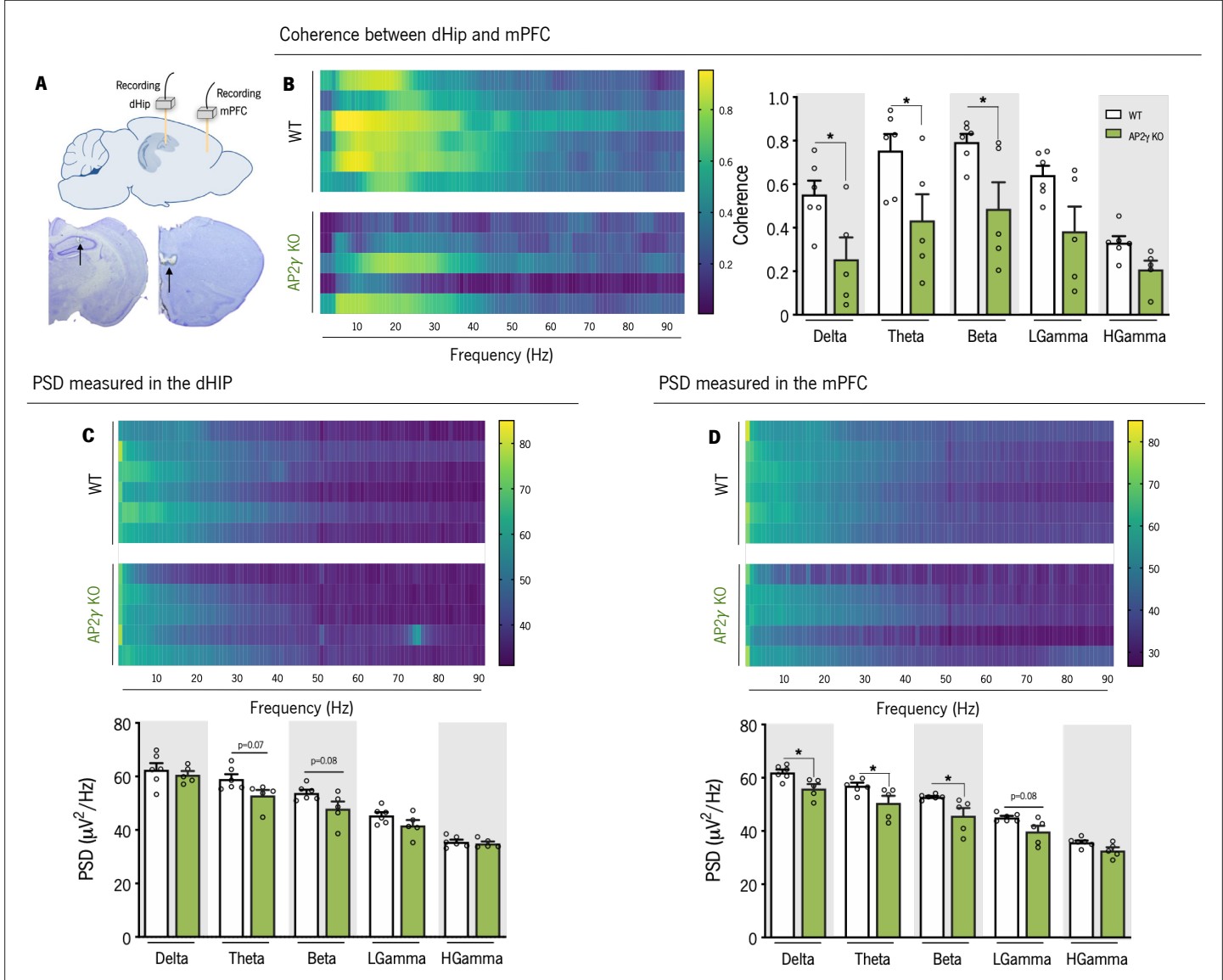

**Figure 6.** In adult mice, AP2γ constitutive and heterozygous deficiency induces deficits in spectral coherence between the dorsal hippocampus (dHip) and the medial prefrontal cortex (mPFC), impacting on neuronal activity. (**A**) Identification of the local field potential (LFP) recording sites, with a depiction of the electrode positions (upper panel), and representative Cresyl violet-stained sections, with arrows indicating electrolytic lesions at the recording sites (lower panel). (**B**) Spectral coherence between the dHip and mPFC (left panel). Group comparison of the coherence values for each frequency (right panel). (**C**) Power spectral density (PSD) was measured in the dHip (**C**) and mPFC (**D**). Heatmaps of PSD activity (upper panel) and group comparison for each frequency (lower panel). Each horizontal line in the Y-axis of the presented spectrograms represents an individual mouse. Frequency bands range: delta (1–4 Hz), theta (4–12 Hz), beta (12–20 Hz), low gamma (20–40 Hz), and High gamma (40–90 Hz). Data presented as mean ± SEM. $n_{WT} = 6$; $n_{AP2γ\ KO} = 5$. [Repeated measures ANOVA; *p < 0.05; Statistical summary in ***Supplementary file 1***]. Abbreviations: WT, wild-type; AP2γ KO, AP2γ heterozygous knockout mice.

The online version of this article includes the following source code and figure supplement(s) for figure 6:

**Source code 1.** Local field potentials analysis between the dorsal hippocampus (dHip) and medial prefontal cortex (mPFC).

**Figure supplement 1.** In adult mice, AP2γ constitutive and heterozygous deficiency does not impact on spectral coherence between the ventral hippocampus (vHip) and the medial prefrontal cortex (mPFC), nor the neuronal activity in each region.

**Figure supplement 1—source code 1.** Local field potentials analysis between the ventral hippocampus (vHip) and medial prefontal cortex (mPFC).

DG of juvenile mice (***Kase et al., 2020***; ***Katsimpardi and Lledo, 2018***), we were unable to analyze their morphology and understand the impact of AP2γ deletion at this period.

Taking into consideration the embryonic and early postnatal developmental modulatory roles of AP2γ, and the severe and/or lethal malformations during development promoted by deficiencies in other members of the AP2 family (AP2α and AP2β) (*Lim et al., 2005*; *Moser et al., 1997*; *Schorle et al., 1996*), we sought to understand whether postnatal hippocampal neurogenesis deficits induced by constitutive and heterozygous deficiency of AP2γ impacts on behavior in specific developmental stages.

The developmental milestones protocol showed no impact of AP2γ constitutive and heterozygous deficiency in early postnatal neurodevelopment. Nevertheless, this deficiency in AP2γ promotes deficits in hippocampal neurogenesis altering the ontogeny of emotional and memory responses in juvenile and adult mice. At juvenile age and during adulthood, AP2γ deficiency led to the manifestation of anxiety-like behavior and significant impairments in different memory behavioral tasks dependent on hippocampal function (*Garthe et al., 2009*; *Garthe and Kempermann, 2013*; *Gu et al., 2012*; *Jessberger et al., 2009*; *Ruediger et al., 2012*; *Treves et al., 2008*). These functional behavioral impairments are either maintained from the juvenile phase until adulthood or emerge in adult mice with deficits in behavioral flexibility and object location recognition together with decreased AHN. Given the importance of AP2γ for the proliferation and expansion of a subpopulation of TAPs (Tbr2⁺), our results reveal that downregulating this specific cell population since early development promotes a significant long-term impact on different behavioral functions. Regulation of TAPs' by AP2γ seems to be also relevant for the preservation of cognitive performance, as shown by its impact on hippocampal-dependent emotional and memory tasks. These observations are in agreement with previous publications where suppression of the TAP's regulator Tbr2 exerted both an anxiety-like phenotype during the juvenile period and induced cognitive deficits during early adulthood (*Veerasammy et al., 2020*). Moreover, Ngn2, another regulator of TAP cells, is also reported as important for the modulation of cognitive behavior, namely in the rescue of cognitive function in the T-Maze task (*Zhao et al., 2018*).

AP2γ deficiency resulted in poor performance on the MWM's behavioral flexibility task and compromised object context discrimination in the adult mice, both of which are memory tasks that rely on the interaction of the hippocampal and prefrontal cortical brain areas (*Barker and Warburton, 2020*; *Hamilton and Brigman, 2015*). Interestingly, adult mice with constitutive and heterozygous deletion of AP2γ present significant deficits of electrophysiological coherence between the dHip and the mPFC in a wide range of frequencies, previously associated to behavior outputs dependent on cortico-limbic networks (*Colgin, 2011*; *Fell and Axmacher, 2011*; *Oliveira et al., 2013*; *Sardinha et al., 2017*). Previously, we observed that conditional deletion of AP2γ during adulthood led to coherence impairments between the vHip-to-mPFC (*Mateus-Pinheiro et al., 2017*). This apparent discrepancy may be related to the specific time of AP2γ deletion on these different mice models. Yet, similar electrophysiological deficits in dHIP-mPFC coherence were observed in a rat model of hippocampal cytogenesis abrogation, which also denote long-term manifestation of emotional and cognitive deficits (*Mateus-Pinheiro et al., 2021*). Moreover, the integrity of the hippocampus-to-PFC circuitry was described to be relevant for the action of antidepressants, such as ketamine (*Carreno et al., 2016*), which promote neurogenesis, suggesting that AP2γ may be involved in conserving this neuronal circuit. Additionally, AP2γ plays an important role on cortical basal progenitors' specification during embryonic development (*Pinto et al., 2009*) that might be affecting the mPFC activity. In fact, AP2γ KO mice presented impaired neuronal activity in the mPFC in a wide range of frequency bands, as detected by the general decrease of PSD signals recorded, and in concordance to previous studies (*Mateus-Pinheiro et al., 2017*). Thus, misspecification of upper cortical layers promoted by AP2γ deficiency since embryonic development may be contributing to the functional electrophysiological alterations, and eliciting cognitive defects herein observed.

Collectively, the findings of this study show that AP2γ transcription factor deficiency compromises the generation of hippocampal glutamatergic neurons, altering the ontogeny of behavior complexity associated with the hippocampus formation. Decreased hippocampal neurogenesis since early development in AP2γ KO mice leads to a compromised basic repertoire of adaptive behaviors. Impairments in DHN induced by deficiencies in AP2γ transcription factor leads to a hypersensitivity toward aversive stimulus as revealed by the presence of an anxious-like phenotype, and memory impairments shown in multiple cognitive tests. These functional behavioral impairments are either maintained from the juvenile phase until adulthood or emerge in adult mice as a result of the reduced AHN caused by constitutive deletion of AP2γ. These observations highlight the importance of an intact hippocampal neurogenic process for such functional outputs.

The current study adds a temporal window of analysis not yet studied and hints that future works should elucidate whether AP2γ can participate in the pathogenesis of neurodevelopmental and/or psychiatric disorders and whether positive modulation of AP2γ could be applied at younger ages as a therapeutic approach to revert deficits in hippocampal neurogenesis and associated behavioral impairments.

# Materials and methods

**Key resources table**

| Reagent type (species) or resource | Designation | Source or reference | Identifiers | Additional information |
|---|---|---|---|---|
| Genetic reagent (*Mus musculus*, male) | *Tfap2c*+/- | Dr. Hubert Schorle (Bonn University Medical School) | | Male mice maintained in a 129/SV background |
| Chemical compound, drug | BrdU, 50 mg/kg | Sigma-Aldrich | # 9,285 | Intraperitoneal injection |
| Antibody | alpha-tubulin (Mouse monoclonal) | Sigma | #5,168 | WB (1:5000) |
| Antibody | Anti-AP2γ (Goat polyclonal) | Abcam | #31,288 | WB (1:500) |
| Antibody | Anti-Pax6 (Rabbit polyclonal) | Millipore | #2,237 RRID:AB_1587367 | WB (1:1000) |
| Antibody | Anti-Sox2 (Mouse monoclonal) | Abcam | #7,935 | WB (1:500) |
| Antibody | Anti-mouse (Goat monoclonal) | BioRad | #1706516 RRID:AB_11125547 | WB (1:10000) |
| Antibody | Anti-rabbit (Goat monoclonal) | BioRad | #1706515 RRID:AB_11125142 | WB (1:10000) |
| Antibody | Anti-goat (Donkey polyclonal) | Santa Cruz Biotechnologies | #sc-2020 RRID:AB_631728 | WB (1:7500) |
| Commercial assay or kit | SuperSignal west Femto reagent | ThermoFisher | #34,096 | |
| Antibody | Anti-BrdU (Rat monoclonal) | Abcam | #6,326 RRID:AB_305426 | IF (1:100) |
| Antibody | Anti-Doublecortin (Rabbit polyclonal) | Abcam | #18,723 RRID:AB_732011 | IF (1:100) |
| Antibody | Anti-Doublecortin (Goat monoclonal) | Santa Cruz Biotechnologies | #sc-8066 RRID:AB_2088494 | IF (1:500) |
| Antibody | Alexa Fluor 488 (Goat Anti-rat polyclonal) | Invitrogen | #32,731 | IF (1:1000) |
| Antibody | Alexa Fluor 568 (Goat Anti-rabbit polyclonal) | Invitrogen | #11,011 RRID:AB_143157 | IF (1:1000) |
| Antibody | Alexa Fluor 594 (Donkey Anti-Goat polyclonal) | Invitrogen | #A32758 RRID:AB_2534105 | IF (1:1000) |
| other | 4',6-diamidino-2-phenylindole (DAPI stain) | Sigma Aldrich | #8,417 | IF (1:200) |
| Commercial assay or kit | Anti-fade Fluorescence Mounting Medium | Abcam | #ab104135 | |
| Software, algorithm | Activity Monitor software | MedAssociates | | |
| Software, algorithm | Kinoscope software | *Kokras et al., 2017* | | |
| Software, algorithm | Fiji software | *Schindelin et al., 2012* | RRID: SCR_002285 | |
| Software, algorithm | Ethovision XT 11.5 | Noldus | RRID:SCR_000441 | |

*Continued on next page*

*Continued*

| Reagent type (species) or resource | Designation | Source or reference | Identifiers | Additional information |
|---|---|---|---|---|
| Software, algorithm | Signal Software | CED | | |
| Software, algorithm | Prism v.8 | GraphPad Software Inc | RRID:SCR_002798 | |
| Software, algorithm | MATLAB | MathWorks Inc | RRID:SCR_005547 | |

## Experimental model details

Wild-type (WT) and AP2γ heterozygous knock-out (*Tfap2c*[+/-], henceforth referred as AP2γ KO) mice were kindly provided by Dr. Hubert Schorle. This previously generated mice line was obtained through a multistep vector- and animal-manipulation (*Werling and Schorle, 2002*). Briefly, a 6 kb genomic fragment spanning from exons 2–7 was used to insert a floxed *neo/tk* cassette 3′ of exon five as well as a *loxP* site 5′ of exon 5. Following electroporation into embryonic stem cells (ES) the correct clones were antibiotically selected using G418. *Cre*-mediated excision removed the region between the *loxP* sites, generating the AP2γ null allele. To generate the AP2γ KO mice, ESs with the null allele were injected into blastocysts resulting in chimeric animals. The offspring of germ line transmitting animals were tested for the presence of the AP2γ null allele. AP2γ KO transgenic mice line was maintained in a 129/SV background and identified by polymerase chain reaction (PCR) of genomic DNA (*Werling and Schorle, 2002*).

Along the study, distinctive cohorts (at least two *per* timepoint) of littermate WT and AP2γ KO male mice were submitted to molecular (n = 4–6 *per* group), behavioral (n = 6–17 *per* group) and electrophysiological (n = 5–6 *per* group) assessment at the different postnatal ages (early postnatal period: from postnatal day (P)one to P21; juvenile period, from P25 to P33; adulthood, from P70 to P84). None of the animals' cohorts were used in multiple timepoints. Different cohorts of animals were used for each analyzed developmental stage to avoid any potential influence of habituation or learning from the repetition of behavior assessment and avoid any long-term impact of behavioral testing during development periods.

As a result of limiting the number of behavioral tests to which each animal was subjected, there is some variation in sample size across behavioral tests. All mice were housed and kept under standard laboratory conditions at 22°C ± 1°C, 55 % humidity, and ad libitum access to food and water on a 12 hr light/dark cycle (lights on 8 A.M. to 8 P.M.). Efforts were made to minimize the number of animals and their suffering. All experimental procedures performed in this work were conducted in accordance with European Regulation (European Union Directive 2010/63/EU) and approved by the Portuguese National Authority for animal experimentation, *Direção-Geral de Alimentação e Veterinária* (DGAV) with the project reference 0420/000/000/2011 (DGAV 4542).

## Behavioral analysis

### Early postnatal period
#### Developmental milestones protocol

Early postnatal neurodevelopment in mice was assessed according to previously validated protocols (*Castelhano-Carlos et al., 2010*; *Guerra-Gomes et al., 2021*; *Hill et al., 2008*; *Santos et al., 2007*). This consisted in a daily evaluation for the first 21 days of life. Mice subjected to early postnatal behavioral assessment did not perform any additional behavioral test later in life. From postnatal day P1 onward, newborn animals were evaluated in several parameters, including skin appearance, activity, and presence of milk spot in the stomach, indicator of correct maternal care and well-being. Pups were examined for the acquisition of developmental milestones until weaning (P21), every day at the same time, in the same experimental room, by the same experimenter. This daily scoring included tests to assess the acquisition of mature response regarding somatic parameters and neurobiological reflexes.

#### Somatic parameters

As a measure of morphological development, animals were daily weighed (weight ±0.01 g). The eye-opening day was also evaluated and considered when both eyes were opened. When both eyes

opened on different days, score was set as one if only one of the eyes was open, and two when both eyes were open. The mature response was registered when both eyes were open.

### Neurobiological reflexes

The assessment of the neurobiological reflexes included the daily performance of different tests. Of note, the scale of evaluation was distinct among tests. Tests including rooting, ear twitch, auditory startle, open field transversal, air righting, wire suspension, postural reflex were scored according to the absence (0) or presence (1) of a mature response. When it was possible to detect a gradual progression in performance as for walking, surface righting, grasping, negative geotaxis, cliff aversion, daily score was attributed between 0 and 3, with 0 representing absence, and three corresponding to the achievement of the mature response. The postnatal day in which animals achieved a mature response was registered. All tests were conducted in a smooth foam pad, and immediately after testing, the pups were returned to their home cage.

## Labyrinthine reflex, body righting mechanism, coordination and strength

*Surface righting reflex* – P1 to P13 – This test consists of gently laying the animal on its back, and the mature response was considered when a pup was able to get right. If the animal did not respond within 30 s, the test was ended. Mature response was achieved when the pups were able to get right in less than 1 s for 3 consecutive days.

*Negative geotaxis* – P1 to P14 – Pups were placed head down in a horizontal grid, tilted 45° to the plane. The acquisition of a mature response was set when pups were able to head up in less than 30 s for 3 consecutive days.

*Air righting* – P8 to P21 – In this test, the pup was held upside down and released from a height of approximately 13 cm from a soft padded surface and released. A mature response was obtained when the animals landed on four paws for 3 consecutive days.

*Cliff aversion* – P1 to P14 – It evaluates the mouse pup's ability to turn and crawl away when on the edge of a cliff. A mature response was achieved once the animal moved away in less than 30 s for 3 consecutive days.

*Postural reflex* – P5 to P21 – Pups were placed in a small plastic box and gently shaken up down and right. When the animals were able to maintain their original position in the box by extending four paws, mature response was acquired.

*Wire suspension* – P5 to P21 – This test evaluates forelimb grasp and strength. Pups were placed vertically to hold with their forepaws a 3 mm diameter metal wire suspended 5 cm above a soft foam pad. A mature response was achieved once the animal was able to grasp the bar, holding it with four paws.

*Grasping* – P5 to P21 – The mouse pup forelimb was touched with a thin wire to evaluate when the involuntary freeing reflex stopped. This reflex disappears with the development of the nervous system, as so, the mature response achieved when the animal grasped immediately and firmly the wire.

## Tactile reflex:

*Ear twitch* – P7 to P15 – In this test, the mouse pup ear was gently stimulated with the tip of a cotton swab, three times. If the animal reacted, flattening the ear against the side of the head for three consecutive days, the mature response was reached.

*Rooting* – P7 to P12 – A fine filament of a cotton swab was used to gently and slowly rub the animal's head, from the front to the back. It was considered a successful test if the pup moved its head toward the filament. The test was repeated on the other side of the head to evaluate the appearance of this neurobiological reflexes on both sides. If the animal did not react to the filament, the test was repeated. Mature response was obtained when animal reacted on both sides for three consecutive days.

## Auditory reflex:

*Auditory startle* – P7 to P18 – We evaluated the reaction of pups to a handclap, at a distance of 10 cm. If pups quick and involuntary jumped for three consecutive days, a matured response was attributed.

## Motor:

*Open field transversal* – P7 to P18 – To execute this test, animals were placed in a small and circle (13 cm diameter), and time to move was recorded. If the pup was not able to move, the test was ended. In case the mouse leaves the circle in less than 30 s, for three consecutive days, a mature response was reached.

*Walking* – P5 to P21 – In this paradigm, animals were able to freely move around for 60 s. The mature response was achieved when they showed a walking movement fully supported on their four limbs.

## Juvenile and adult mice

During juvenile and adult periods and through different previously validated behavior tests, we assessed sequentially anxiety-like behavior, coping and anhedonic behavior, and the performance on multiple cognitive tasks. Individual cohorts of mice were exclusively subjected to behavioral characterization at one of these periods, either while juvenile or later at adulthood. To avoid any possible impact of habituation or learning from the repetition of behavior assessment, we exposed different cohorts of animals to a subset of behavioral paradigms. Thus, with this approach of limiting the number of behavioral tests each animal was exposed to, there is some variation in the sample number across behavioral paradigms. Behavioral tests were performed in the following order: Open-field, elevated-plus maze, novelty-suppressed feeding, sucrose splash, forced-swimming, tail suspension, object recognition, pattern separation, Morris water maze, and contextual fear conditioning. When mice were exposed to multiple behavioral paradigms, 1–2 days of resting were added between tests. Sucrose splash test was only performed in juvenile mice while elevated-plus maze and forced-swimming test were exclusively tested during the adult period.

## Open field (OF) test

The OF test is a behavioral paradigm commonly used to assess anxiety-like and exploratory behavior, and general activity in rodents. This apparatus consists of a square arena of 43.2 cm x 43.2 cm, closed by a 30.5 cm high wall, with a lamp at the center (~850 lux). Mice were individually placed in the center of the OF arena, and their movement was tracked for 5 mins, using a 16-beam infrared system (MedAssociates, US). Average velocity of the animals was considered as a measure of locomotor capacity, and the percentage of distance traveled in the center of the arena (defined as the central 11 × 11 cm region) was considered as measurement of anxiety-like behavior. Data was analysed using the Activity Monitor software (MedAssociates, US). For juvenile behavioral assessment, mice were exposed to this paradigm at P25, and adult animals were between P70 and P84.

## Elevated-plus maze (EPM) test

To assess anxiety-like behavior, the EPM test was also performed (***Walf and Frye, 2007***). This test is performed on a black propylene apparatus (ENV – 560; MedAssociates Inc, US) with two opposite open arms (50.8 cm x 10.2 cm) and two closed arms (50.8 cm x 10.2 cm x 40.6 cm) elevated 72.4 cm above the floor and dimly illuminated. The central area connecting both arms corresponds to an area of 10 × 10 cm. Animals were individually positioned in the center of the maze, facing an edge of a closed-arm, and were allowed to freely explore the maze for 5 min. All trials were recorded using an infrared photobeam system, and the percentage of time spent in the open arms was accessed through the EthoVision XT 11.5 tracking system (Ethovision, Noldus Information Technologies, Netherlands). For the adult behavioral assessment, animals were between P70 and P84.

## Novelty suppressed feeding (NSF) test

Following ~18 h of food deprivation, animals were placed in an illuminated open-field arena with 43.2 cm x 43.2 cm, closed by a 30.5 cm high wall, with a lamp at the center (~850 lux). The arena floor was covered with 2 cm of the habitual bedding material, and a single food pellet was placed on a 5 cm square piece of filter paper at the center. Mice were placed facing a wall, and the latency to reach and feed on the food pellet was assessed, within a maximum limit of 10 min. Feeding behavior was defined as rearing with visible food consumption. Latency to reach and feed on the food pellet was used as a measure of anxiety-like behavior (***Samuels and Hen, 2011***). After this test trial, animals were single housed in homecages containing a pre-weighted food pellet, and allowed to feed during

10 min. The amount of food consumption in the home cage provided a measure of appetite drive (**Bessa et al., 2009**). For juvenile behavioral assessment, mice were exposed to this paradigm at P27, and adult animals were between P70 and P84.

## Forced swimming test (FST)

For the assessment of adaptive behavior to cope with an inescapable stressor, we performed the forced swimming test (**Porsolt et al., 1977**). Briefly, each animal was individually placed in glass cylinders filled with water (23 °C; depth 30 cm) for 5 min. All sessions were video-recorded, and the immobility time, defined through a video tracking software Ethovision XT 11.5 (Noldus, Netherlands), was considered as a measure of coping behavior. Mice were considered immobile when all active behaviors (struggling, swimming, and jumping) were ceased. For immobility, the animals had to remain passively floating or making minimal movements required to maintain the nostrils above water. For the assessment of coping behavior, the first 3 min of the trial were considered as a habituation period and the last 2 min as the test period (**Mateus-Pinheiro et al., 2017**). For the adult behavioral assessment, animals were between P70 and P84.

## Tail-suspension test (TST)

The TST is a commonly used behavioral test to assess adaptive behavior to cope with an inescapable stressor in rodents. Briefly, mice were suspended by the tail to the edge of a laboratory bench 80 cm above the floor (using adhesive tape) for 6 min. Trials were video-recorded, and the immobility and climbing times were automatically analyzed by the video tracking software Ethovision XT 11.5 (Noldus, Netherlands). In the TST, immobility time is normally defined as a readout of coping behavior. Animal models of depression typically exhibit longer periods of immobility during the TST (**Krishnan and Nestler, 2011**). For the juvenile behavioral assessment, mice were exposed to this paradigm at P26, and adult animals were between P70 and P84.

## Splash-sucrose test (ST)

The splash-sucrose test consists of spraying a 10 % sucrose solution on the dorsal coat of mice in their home cage (**Yalcin et al., 2008**). Sucrose solution in the mice's coat induces grooming behavior. After spraying the animals, animal's behavior was video-recorded for 5 min and the time spent grooming was taken as an index of self-care and motivational behavior. Grooming was manually quantified using the behavioral scoring program Kinoscope (**Kokras et al., 2017**). For juvenile behavioral assessment, mice were exposed to this paradigm at P27.

## Object recognition test (ORT)

The ORT was performed to assess short- and long-term memory (**Leger et al., 2013**). This test relies on rodents' nature to explore and preference for novelty. First, mice were acclimatized to a testing arena (30 cm x 30 cm x 30 cm) under dim light (desk lamp pointing to a room wall ~150–200 lux at the arenas level) for 3 days during 20 min. After habituation, animals were presented with two equal objects for 10 min (training), positioned in the center of the arena. One hour later, one of the objects was moved toward an arena wall, and mice were allowed to freely explore the objects for 10 min. On the following day, animals returned to the arena for 10 min, with one of the objects being replaced by a novel object. The familiar and novel objects had different size, color, shape, and texture. These objects were made from building blocks as represented in **Figure 3G**. Between trials, the arena and objects were properly cleaned with 10 % ethanol. Sessions were recorded and manually scored through the behavioral scoring program Kinoscope (**Kokras et al., 2017**). The percentage of time exploring the moved- and the novel-object was used as a measure of short- and long-term memory, respectively. For juvenile behavioral assessment, mice were exposed to this paradigm between P27 – P31, and adult animals were between P70 – P84.

## *Object-in-Context Recognition Test* (OIC)

The OIC test is used to measure animal's ability to distinguish between similar events (**Dere et al., 2007**; **Jain et al., 2012**; **Ramsaran et al., 2016**). This behavioral paradigm was performed in white and grey arenas (30 cm x 30 cm x 30 cm) under dim white-light room illumination (desk lamp pointing to a room wall ~150–200 lux at the arenas levels). During the training phase, animals were placed in an open white chamber wall-patterned with laminated blue squares and allowed to explore two identical

objects for 10 min. After 30 min inter-trial interval, mice were placed in a second open grey chamber wall-patterned with laminated white stripes. Animals were allowed to explore for 10 min two identical objects unique from the first trial. Both set of objects were made from building blocks and had distinct surface pattern (one set was smooth and the other softly wrinkled). The objects used in this test had a similar shape to the representation in *Figure 3J*. After 3 hr, mice were exposed for 10 min to the test trial. Mice from both experimental groups were randomly assigned to either context A or B. During the test trial, one of the objects presented in the sample trials A and B is also presented. The time that each mouse spent exploring the out of context object was compared with the time exploring the object in the familiar context (e.g. object from trial A in context from trial B). Between trials, the arena and objects were properly cleaned with 10 % ethanol. The test session was recorded and manually scored through the behavioral scoring program Kinoscope (*Kokras et al., 2017*). For juvenile behavioral assessment, mice were exposed to this paradigm at P29, and adult animals were between P70 and P84.

## Contextual-fear conditioning (CFC)

The CFC test was performed in a white acrylic box with internal dimensions of 20 cm wide, 16 cm deep, and 20.5 cm high (MedAssociates). CFC apparatus had a fixed light bulb mounted directly above the chamber to provide a source of illumination. Each box contained a stainless-steel shock grid floor inside a clear acrylic cylinder (10 cm diameter with 15 cm depth), where the animals were placed. All animals were exposed to two probes: a context probe and a cue (light) probe, as previously described (*Gu et al., 2012*; *Mateus-Pinheiro et al., 2017*). All probes were recorded, and the freezing behavior was manually scored through Kinoscope (*Kokras et al., 2017*). For juvenile behavioral assessment, mice were exposed to this paradigm between P31 and P33, and adult animals were between P70 and P84. This behavioral paradigm took 3 days.

### Day 1:

Animals were individually placed in the conditioning-white box (Context A) and received three pairings between a light (20 s) and a co-terminating shock (1 s, » 0.5 mA). The interval between pairings was 180 s, and the first light presentation started 180 s after the beginning of the trial. After the three pairings, mice remained in the acrylic box for 30 s, being after returned to their home cage. Between animals, the apparatus was properly cleaned with 10 % ethanol.

### Day 2:

For the context probe, animals were placed into the same white acrylic chamber (context A), 24 hr after the light-shock pairings. The freezing behavior was monitored for 3 min. Two hours later, animals were introduced into a modified version of the chamber (Context B). This new box was sheeted with a black plasticized cover, sprayed with a vanilla scent. In this way, both contexts had distinct spatial and odor cues. Also in Context B, the ventilation was off, and the experimenter wore a different color of gloves and a lab coat. Freezing behavior was measured for 3 min. The freezing behavior state was defined as the total absence of motion, for a minimum of 1 s.

### Day 3:

For the cue probe, the animals were set in Context B, and individually placed in this chamber 24 hr after the context probe. After 3 min, the light was turned on for 20 s, and the freezing behavior monitored for 1 min after light is turned off.

## Morris water maze (MWM)

In the MWM test, several cognitive domains were assessed: working- and spatial-reference memory and behavioral flexibility. Additionally, the strategies used to reach the platform were also analyzed. MWM was performed in a circular white pool (170 cm diameter) filled with water at 22 °C to a depth of 31 cm in a room with and dim light and extrinsic clues (triangle, square, cross, and horizontal stripes). The pool was divided into four quadrants by imaginary lines, and a clear-acrylic cylinder platform (12 cm diameter; 30 cm high), placed in one of the quadrants. All trials were video recorded by a tracking system (Viewpoint, France). For juvenile behavioral assessment, mice were exposed to this paradigm between P29 and P32 and adult animals were between P70 and P84.

### Working memory task

The working memory task (*Alves et al., 2017*; *Cerqueira et al., 2007*) evaluates the cognitive domain that relies on the interplay between the hippocampal and prefrontal cortex (PFC) functions. In this task, animals had to learn the position of the hidden platform and to retain this information for four consecutive daily trials. The task was performed for 4 days and in a clockwise manner the platform was repositioned in a new quadrant each day. During the daily trials, animals had different starting positions (north, east, west, and south). Trials ended when mice reached the platform within the time limit of 120 s. If the animals did not reach the platform during the trial time, they were guided to the platform and allowed to stay for 30 s. The time and path to reach the platform were recorded.

### Reference memory task

After working memory evaluation (days 1-4), spatial-reference memory, a hippocampal dependent function, was assessed by keeping the platform in the same quadrant during three consecutive days (days 4-6) (*Morris, 1984*). During the daily trials, animals had different starting positions (north, east, west, and south). The time and path to reach the platform were recorded for each trial.

### Reversal learning task:

On the last day of MWM testing, reversal-learning performance, a PFC dependent function, was assessed. This was conducted by positioning the platform in a new (opposite) quadrant. Animals were tested in 4 trials. The percentage of time spent in the new and old quadrant containing the platform was used as readout of behavioral flexibility.

### Search strategies analysis:

Throughout the Morris water maze, the mice adopted strategies to reach the hidden platform were evaluated as previously described (*Garthe and Kempermann, 2013*; *Mateus-Pinheiro et al., 2017*; *Ruediger et al., 2012*). Quantitative analyses and strategy classification were completed by assessing different parameters collected through the Viewpoint software: (1) thigmotaxis (Tt): most of the swim distance (>70%) happened within the outer ring area (8 cm from the pool border; (2) random swim (RS): most of the swim distance (>80%) occurred within the inner circular area, and all quadrants were explored with a percentage of swim distance not below 50% for none of the quadrants; non-circular trajectories; (3) scanning (Sc): most of the swim pattern and distance (>80%) happened within the inner circular area, with balanced exploration in all quadrants of the pool; non-circular trajectories, with a percentage (80%), with a balanced exploration of all pool quadrants; swim distance in the plat-form corridor area 80%); swim distance in the platform corridor area >60%, with shifts in the trajectories directions; (6) focal search (FS): directed trajectories to the platform zone, with swim exploration within the perimeter of the escape platform (30cm); (7) directed swim (DSw): directed trajectories to the hidden platform, without much exploration of the pool. For simplification, we defined two blocks of strategies: Block 1, that comprises the 'non-hippocampal dependent strategies' (Tt, RS, and Sc), and Block 2, comprising the defined 'hippocampal dependent strategies' (DS, FS, and DSw). These blocks were defined when a sequence of at least three trials within the same block were reached.

## Electrophysiological studies

Electrophysiological recordings were obtained from anesthetized mice (sevoflurane 2.5%; 800 mL/min). Surgical procedure was performed to insert platinum/iridium concentric electrodes (Science Products) in the target positions following the mouse brain atlas (from Paxinos): prelimbic region of the medial prefrontal cortex (mPFC): 1.94 mm anterior to bregma, 0.4 mm lateral to the midline, 2.5 mm below bregma; dorsal hippocampus (dHIP): 1.94 mm posterior to bregma, 1.2 mm lateral to the midline, 1.35 mm below bregma; ventral hippocampus (vHIP): 3.8 mm posterior to bregma, 3.3 mm lateral to the midline, 3.4 mm below bregma. LFP signals obtained from mPFC, dHIP, and vHIP were amplified, filtered (0.1–300 Hz, LP511 Grass Amplifier, Astro-Med), acquired (Micro 1401 mkII, CED) and recorded through the Signal Software (CED). Local field activity was recorded at the sampling rate of 1000 Hz during 100s. After electrophysiological recordings, a biphasic 0.7 mA stimulus was delivered to mark the recording sites. Then, mice were deeply anesthetized with sodium pentobarbital, brains removed, immersed in paraformaldehyde (PFA) 4% for 48 hr and sectioned (50 µm) in a vibratome (Leica VT 1000S, Germany). Coronal slices containing the mPFC, dHip vHip were stained for Cresyl Violet to check for recording sites. Animals with recording positions outside at least in one

of the two regions under study (mPFC and dHip or vHip) were excluded from the analysis. Coherence analysis was based on multi-taper Fourier analysis.

Coherence was calculated by custom-written MATLAB scripts, using the MATLAB toolbox Chronux (http://www.chronux.org; *Figure 6—source code 1*: Local field potentials analysis between the dHip and mPFC; and *Figure 6—figure supplement 1—source code 1*: Local field potentials analysis between the vHip and mPFC) (*Mitra and Pesaran, 1999*). Coherence was calculated for each 1 s long segments and their mean was evaluated for all frequencies from 1 to 90 Hz. The power spectral density (PSD) of each channel was calculated through the 10 x log of the multiplication between the complex Fourier Transform of each 1s long data segment and its complex conjugate. The mean PSD of each channel was evaluated for all frequencies from 1 to 90 Hz (*Oliveira et al., 2013*). Both coherence and PSD measurements were assessed in the following frequencies: delta (1–4 Hz), theta (4–12 Hz), beta (12–20 Hz); low (20–40 Hz), and high gamma (40-90 Hz). BrdU labeling: to assess proliferation of fast-dividing progenitor cells, and its impact on the generation of adult-born neurons, animals from all groups were injected intraperitoneally with the thymidine analogous 5-bromo-2'-deoxyuridine or bromodeoxyuridine (BrdU, 50 mg/kg; Sigma-Aldrich, US) that is incorporated in the DNA during the S-phase. BrdU injections were performed once, at the end of the behavioral assessment, 24 hr prior to occision.

## Western blot analysis

Hippocampal DG of juvenile and adult AP2γ KO mice and WT littermates were carefully macrodissected out after occision. The tissue was weighted and homogenized in RIPA buffer [containing 50 mM Tris HCl, 2 mM EDTA, 250 mM NaCl, 10 % glycerol, 1 mM PMSF protease inhibitors (Roche, Switzerland)] and then sonicated (Sonics & Materials, US) for 2 min. Samples were centrifuged for 25 min at 10,000 rpm and 4 °C. The protein concentration of the supernatant was determined using Bradford assay. Samples with equal amounts of protein, 30 μg, were analyzed using the following primary antibodies: alpha-tubulin (#5168; Sigma, mouse, 1:5000), AP2γ (#31288; goat, 1:500; Abcam, UK), Pax6 (#2237; rabbit, 1:1000; Millipore, US), Sox2 (#7935; mouse, 1:500; Abcam, UK) and Tbr2 (#2283; rabbit, 1:500; Millipore, US). Secondary antibodies were used from BioRad (Anti-mouse, 1:10.000; #1706516; Anti-rabbit, 1:10.000; #1706515; US) and Santa-Cruz Biotechnologies (Anti-goat, 1:7500; #A2216, US). Membranes were developed using SuperSignal west Femto reagent (#34096; ThermoFisher, US) and developed in Sapphire Biomolecular Imager from Azure Biosystems (US). After developing, images were quantified using AzureSpot analysis software (Azure Biosystems, US).

## Immunostaining procedures

All mice were deeply anesthetized and then transcardially perfused with cold 0.9 % NaCl, followed by 4 % paraformaldehyde (PFA). Brains were carefully removed from the skull, postfixed in 4 % PFA, and then cryoprotected in 30 % sucrose solution. The brains were coronally sectioned in the vibratome (Leica VT 1000 S, Germany) with a thickness of 50 mm, extending over the entire length of the hippocampal formation. Coronal sections containing the dorsal hippocampal dentate gyrus (DG) were further stained to assess cell proliferation, quantify the population of neuroblasts, immature neurons and its morphology. Analyses were focused in the dorsal hippocampal DG due to the results obtained in the electrophysiological studies.

For the cell proliferation and neuroblasts population assessment, brain sections containing the dorsal hippocampal DG were double stained for BrdU (#6326; rat, 1:100; Abcam, UK) and double-cortin (DCX; #18723; rabbit, 1:100; Abcam, UK). Appropriate secondary fluorescent antibodies were used (Alexa Fluor 488 Goat Anti-rat, #32731; 1:1000; Invitrogen, US; and Alexa Fluor 568 Goat Anti-rabbit, #11011; 1:1000; Invitrogen, US). For cell nuclei labeling, 4',6-diamidino-2-phenylindole (DAPI, 1:200; Sigma Aldrich) was used. The density of each cell population in the DG was determined by normalizing the number of positive cells by the corresponding area of the region.

For the morphology analysis of immature neurons, brain sections containing the dorsal hippocampal DG were stained for DCX (#sc-8066; goat, 1:500; Santa Cruz Biotechnology, US) and the corresponding secondary antibody was used (Alexa Fluor 594 donkey anti-goat, #A32758, 1:1000; Invitrogen, US), as previously described (*Dioli et al., 2019*). For cell nuclei labeling, 4',6-diamidino-2-phenylindole (DAPI, 1:200; Sigma Aldrich) was used.

All sections were mounted with anti-fade fluorescence mounting medium (Fluorshield Mounting Medium, #ab104135, Abcam, UK). Analysis and cell counting were performed using a confocal microscope (Olympus FluoViewTM FV1000, Germany) and an optical microscope (Olympus BX51, Germany). The observer was blind to the experimental groups.

## Morphological analysis

Dendritic reconstruction of DCX$^+$ cells was performed by the analysis of confocal stack images (Olympus FluoViewTM FV1000, Germany) by the simple neurite tracer plugin (*Dioli et al., 2019*; *Longair et al., 2011*; *Schindelin et al., 2012*) using the open-access Fiji software, as previously described (*Dioli et al., 2019*; *Valero et al., 2014*). For a better dendrite segmentation, the colored confocal images were converted to black and grey. In this study, only DCX$^+$ cells that branch into the granular cell layer (GCL) and reach the molecular layer (ML) were studied. Sholl analysis was also assessed using for that the 'Sholl analysis plugin (http://fiji.sc/Sholl_Analysis)' and based on the quantification of the number of intersections between dendrites and the surface of circles with a radius increment of 20 μm. The dendritic length and the neuronal complexity were analyzed with the Skeletonize (https://imagej.net/Skeletonize3D) and AnalyzeSkeleton (https://imagej.net/AnalyzeSkeleton). The experimenter was blind to the animal's genotype.

3D dendritic morphology of pre-existing granule neurons in the DG of juvenile and adult mice was performed through the Golgi-Cox impregnation technique. Briefly, brains were immersed in Golgi-Cox solution for 14 days and then transferred to 30 % sucrose solution. Coronal sections (200 μm) were sectioned in a vibratome (Leica VT100S, Germany), collected and then blotted dry onto gelatine-coated microscope slides. Sections containing the dorsal hippocampus were then alkalinized in 18.8 % ammonia, developed in Dektol (Kodak, US), fixed in Kodak Rapid Fix, dehydrated and xylene cleared. Dendritic arborization was analyzed in the DG of WT and AP2γ KO animals (10 neurons *per* animal).

## Data analysis and statistics

Statistical analysis was performed using Prism v.8 (GraphPad Software, US). Animals were randomly assigned to groups, balanced by genotypes. Sample sizes were determined by power analyses based on previously published studies (*Mateus-Pinheiro et al., 2017*) and normal distributions were assessed using the Shapiro-Wilk statistical test, taking into account the respective histograms and measures of skewness and kurtosis. For variables that followed the Gaussian distribution within groups, parametric tests were applied, while non-parametric tests were used for discrete variables. To compare the mean values for two groups, a two-tailed independent-sample t-test was applied. For comparisons between two time-points a two-way ANOVA was used. For longitudinal analyses (across days and different trials) a repeated measures ANOVA was used.

For the comparison of categorical variables (strength to grab, limb grasping, and clasping), cross-tabulations were performed and the statistical test used was Fisher's exact test (when Pearson Qui-Squared assumptions were not met).

Data is expressed either as mean ± SEM (standard error of the mean), as median, or as percentage, as stated in the figures' legends. Statistical significance was set when $p < 0.05$. Statistical summary presented in *Supplementary file 1*.

## Acknowledgements

We would like to acknowledge Dr. Hubert Schorle for providing the AP2γ mice line. ELC, AMP, PP, CSC, JS, TSR, BMP, AVD, JFO, NDA, and LP received fellowships from the Portuguese Foundation for Science and Technology (FCT) (IF/00328/2015 to JFO; 2020.02855.CEECIND to LP). This work was funded by FCT (IF/01079/2014, PTDC/MED-NEU/31417/2017 Grant to JFO), BIAL Foundation Grants (037/18 to JFO and 427/14 to LP), "la Caixa" Foundation (ID 100010434 to AJR), under the agreement LCF/PR/HR20/52400020), the European Research Council (ERC) under the European Union's Horizon 2020 research and innovation program (grant agreement No 101003187 to AJR), and Nature Research Award for Driving Global Impact - 2019 Brain Sciences (to LP). This was also co-funded by the Life and Health Sciences Research Institute (ICVS), and by FEDER, through the Competitiveness Internationalization Operational Program (POCI), and by National funds, through the Foundation for Science and Technology (FCT) - project UIDB/50026/2020 and UIDP/50026/2020. Moreover, this work has been funded by ICVS Scientific Microscopy Platform, member of the national infrastructure PPBI

- Portuguese Platform of Bioimaging PPBI-POCI-01–0145-FEDER-022122; by National funds, through the Foundation for Science and Technology (FCT) - project UIDB/50026/2020 and UIDP/50026/2020.

## Additional information

### Funding

| Funder | Grant reference number | Author |
|---|---|---|
| Fundação para a Ciência e a Tecnologia | SFRH/BD/131278/2017 | Eduardo Loureiro-Campos |
| Fundação para a Ciência e a Tecnologia | SRFH/BD/120124/2016 | Bárbara Mendes-Pinheiro |
| Fundação para a Ciência e a Tecnologia | SFRH/BD/135273/2017 | Tiago Silveira-Rosa |
| Fundação para a Ciência e a Tecnologia | SFRH/BD/147066/2019 | Ana Verónica Domingues |
| Fundação para a Ciência e a Tecnologia | CEECIND/03887/2017 | Carina Soares-Cunha |
| Fundação para a Ciência e a Tecnologia | IF/00328/2015 | João Oliveira |
| Fundação para a Ciência e a Tecnologia | 2020.02855.CEECIND | Luísa Pinto |
| Fundação Bial | 037/18 | João Oliveira |
| Fundação Bial | 427/14 | Luísa Pinto |
| Fundação para a Ciência e a Tecnologia | UIDB/50026/2020 and UIDP/50026/2020 | Eduardo Loureiro-Campos<br>António Mateus-Pinheiro<br>Joana Silva<br>Vanessa Morais Sardinha<br>Bárbara Mendes-Pinheiro<br>Tiago Silveira-Rosa<br>Ana Verónica Domingues<br>Ana João Rodrigues<br>João Oliveira<br>Nuno Sousa<br>Luísa Pinto |
| Fundação para a Ciência e a Tecnologia | PPBI-POCI-01-0145-FEDER-022122 | Eduardo Loureiro-Campos<br>António Mateus-Pinheiro<br>Joana Silva<br>Vanessa Morais Sardinha<br>Bárbara Mendes-Pinheiro<br>Tiago Silveira-Rosa<br>Ana Verónica Domingues<br>Ana João Rodrigues<br>João Oliveira<br>Nuno Sousa<br>Luísa Pinto |
| Fundação para a Ciência e a Tecnologia | IF/01079/2014 | João Oliveira |
| Fundação para a Ciência e a Tecnologia | PTDC/MED-NEU/31417/2017 | João Oliveira |
| La Caixa Foundation (LCF/PR/HR20/52400020) | 100010434 | Ana João Rodrigues |
| European Research Council | 10100 3187 | Ana João Rodrigues |

The funders had no role in study design, data collection and interpretation, or the decision to submit the work for publication.

## Author contributions
Eduardo Loureiro-Campos, Conceptualization, Data curation, Formal analysis, Investigation, Methodology, Validation, Visualization, Writing – original draft, Writing – review and editing; António Mateus-Pinheiro, Formal analysis, Methodology; Patrícia Patrício, Formal analysis, Methodology, Writing – review and editing; Carina Soares-Cunha, Investigation, Methodology, Writing – review and editing; Joana Silva, Tiago Silveira-Rosa, Investigation, Methodology; Vanessa Morais Sardinha, Bárbara Mendes-Pinheiro, Formal analysis, Investigation, Methodology; Ana Verónica Domingues, Methodology; Ana João Rodrigues, Methodology, Resources, Writing – review and editing; João Oliveira, Formal analysis, Methodology, Resources, Writing – review and editing; Nuno Sousa, Formal analysis, Funding acquisition, Methodology, Resources, Writing – review and editing; Nuno Dinis Alves, Conceptualization, Data curation, Formal analysis, Funding acquisition, Investigation, Methodology, Resources, Validation, Visualization, Writing – original draft, Writing – review and editing; Luísa Pinto, Conceptualization, Data curation, Formal analysis, Funding acquisition, Investigation, Methodology, Project administration, Resources, Supervision, Validation, Visualization, Writing – original draft, Writing – review and editing

## Author ORCIDs
Eduardo Loureiro-Campos (iD) http://orcid.org/0000-0001-5834-5851
Tiago Silveira-Rosa (iD) http://orcid.org/0000-0001-6159-1623
Ana João Rodrigues (iD) http://orcid.org/0000-0003-1968-7968
João Oliveira (iD) http://orcid.org/0000-0002-1005-2328
Nuno Dinis Alves (iD) http://orcid.org/0000-0002-8712-3710
Luísa Pinto (iD) http://orcid.org/0000-0002-7724-0446

## Ethics
Efforts were made to minimize the number of animals and their suffering. All experimental procedures performed in this work were conducted in accordance with the EU Directive 2010/63/EU and approved by the Portuguese National Authority for animal experimentation, Direção-Geral de Alimentação e Veterinária (DGAV) with the project reference 0420/000/000/2011 (DGAV 4542).

## Decision letter and Author response
Decision letter https://doi.org/10.7554/eLife.70685.sa1
Author response https://doi.org/10.7554/eLife.70685.sa2

# Additional files

## Supplementary files
• Supplementary file 1. Statistical summary of results.
• Transparent reporting form

## Data availability
All data generated or analysed during this study are included in the manuscript and supporting file; Source Data file has been provided for Figure 1. Source Code files have been provided for Figure 6 and for Figure 6 - Figure supplement 1.

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
