## [Editor Report]

The aim of this study was to examine the impact of AP2γ deficiency on the development of sensorimotor skills, cognitive, and emotional function. This paper will be of interest to scientists in the fields of developmental psychobiology and neurogenesis.

---

## [Decision Letter]

**Decision letter after peer review:**

Thank you for sending your article entitled "Constitutive AP2γ deficiency reduces postnatal hippocampal neurogenesis and induces behavioral deficits in juvenile mice that persist during adulthood" for peer review at *eLife*. Your article has been reviewed by 3 peer reviewers, one of whom is a member of our Board of Reviewing Editors, and the evaluation has been overseen by Catherine Dulac as the Senior Editor.

All reviewers found the paper interesting for scientists in the fields of developmental psychobiology and molecular mechanisms underlying adult hippocampal neurogenesis. The data analysis is rigorous, and the conclusions are justified by the data.

However, additional data from the different ontogenic stages would be helpful to strength the conclusions. In addition, given the authors previous publication on the impact of conditional knockdown of AP2γ (Mateus-Pinheiro et al., Mol Psy 2017), the authors need to emphasise what the novelty of this work is and how it advances their previous findings.

In addition, more information is needed to improve the manuscript. In particular, we recommend to better describe the animal model, the methods and use anova for the analysis of coherence or PSD at different frequencies.

*Reviewer #1:*

The aim of this study is to examine the impact of AP2γ deficiency on the emergence of behavioral impairments of emotional and cognitive functions. This longitudinal behavioral analysis show that AP2γ deficiency anxiety leads to memory impairments and an anxious-like phenotype during the juvenile period, a phenotype that persists into adulthood. In contrast, depression-like responses remain unchanged. These changes are associated with an alteration in hippocampal neurogenesis and an alteration in adult hippocampal-to-PFC functional connectivity.

Major strength:

Understanding when emotional and cognitive functions emerge during development is a fundamental question with regard to neurodevelopmental disorders. In this study, the authors addressed this question by determining the impact of AP2γ deficiency on the emergence of different behavioral responses. They found some neurobiological correlates for the deficits observed.

Major weakness:

It is unclear whether different batches of animals were used for the behavioral testing or alternatively whether the same batch of animals was tested during two different developmental periods. In this latter case, the impact of training during the juvenile period may have interfered with the behavior of animals tested at adulthood. In addition, it would have been interesting to test (1) anxiety before weaning, (2) spatial learning in a WM or contextual fear conditioning in juvenile/adolescent animals.

Given that the behavioral syndrome is a stable form from the juvenile period into adulthood, the role of adult hippocampal neurogenesis in this context is unclear.

The authors found no differences when assessing DG neuronal characteristics in juvenile and adult mice. It would be interesting to determine whether AP2γ deficiency impacts the dendritic arborisation of Dcx neurons in juvenile and adult mice.

*Reviewer #2:*

Loureiro-Campos, Dinis Alves et al. investigated impact of a constitutive AP2γ; heterozygous deletion in mice from early postnatal development until adulthood on hippocampal neurogenesis, emotional and cognitive behaviors, and limbic-cortical connectivity. The authors demonstrate that constitutive and heterozygous AP2γ deficiency does not alter early postnatal development but reduces BrdU-positive and BrdU/DCX double-positive cells, but not dendritic length, in the dentate gyrus of the hippocampus both at juvenile and adult periods. The authors also find that constitutive and heterozygous AP2γ deficiency in juvenile mice (P25-31) increases anxiety-like behavior (% distance center zone open-field test), without altering coping behavior in response to an acute stressor (immobility during tail suspension test) or depressive-like behavior (grooming during sucrose splash test), while promoting cognitive deficits (novel object recognition during object recognition test). Constitutive and heterozygous AP2γ deficiency in adult mice (P70-92) increases anxiety-like behavior (% distance center zone open-field test, % time in open arms elevated plus-maze), without altering coping behavior in response to an acute stressor (immobility during forced swim test or tail suspension test). Adult AP2γ+/- also showed altered behavior during the object recognition test (trend for decreased preference to explore novel object location) and deficits in contextual hippocampal-related memory (reduced freezing time during exposure to familiar context during the contextual fear conditioning test). Adult AP2γ+/- also showed impaired behavioral flexibility (reference memory test in Morris water maze) and a delayed switch from non-hippocampal-dependent to hippocampal-dependent strategies to reach the escape platform during consecutive Morris water maze trials. Finally, simultaneous assessment of electrophysiological features of local field potentials (LFPs) in the dorsal hippocampus and medial prefrontal cortex revealed an impaired functional connectivity between these two brain regions in adult AP2γ+/- mice compared to wild-type controls, and decreases in power spectral density, predominantly in the medial prefrontal cortex.

The conclusions of this paper are supported by the data. The manuscript is written clearly, and the figures are organized and informative. The Material and Methods section is very detailed.

Adding individual datapoints to appropriate graphs will increase clarity for the reader. The rationale behind the current study could be described in greater detail; it could, for example, be better explained why constitutive and heterozygous AP2;; deficiency was studied in this paper. A reader would also benefit from additional information how AP2γ+/- mice were exactly generated (e.g. compare with detailed information on experimental animals in Mateus-Pinheiro et al. 2017), and what the information form this paper specifically adds to previous findings form the same group with (mostly) constitutive and homozygous AP2γ deficiency in mice (Mateus-Pinheiro et al. 2017).

The data of this paper add to the knowledge on the role of AP2γ deficiency in mice on hippocampal neurogenesis, emotional and cognitive behaviors, and limbic-cortical connectivity.

Comments for the authors:

I have the following recommendations to improve the clarity of the manuscript.

– Adding individual datapoints to the graph would increase clarity for the reader.

– The rationale behind the current study could be described in greater detail; it could, for example, be better explained why constitutive and heterozygous AP2γ deficiency was studied in this paper.

– The reader would also benefit from detailed information how these mice were exactly generated (e.g. see information on Animals in Mateus-Pinheiro et al. 2017).

– The reader would also benefit what the information from this paper specifically adds to previous findings form the same group with (mostly) constitutive and homozygous AP2γdeficiency in mice (Mateus-Pinheiro et al. 2017)

– The repeated analysis of coherence or PSD at different frequencies within samples with simple Student t-tests feels incorrect. A repeated measures ANOVA might be more appropriate. At the minimum, a multiple comparison correction should be applied.

*Reviewer #3:*

The data presented build upon the authors previous findings reporting that conditional knockout of AP2γ in adulthood decreases hippocampal neurogenesis and interferes with cognitive performance. Here, the authors report that these impairments also occur following constitutive AP2γ knockdown and are apparent prior to adulthood, namely in the juvenile period. The authors also report that constitutive AP2γ knockdown disrupts dorsal hippocampus-medial prefrontal cortex (mPFC) coherence but does not affect ventral hippocampus-mPFC coherence which opposes the authors previous findings when investigating the impact of conditional AP2γ knockout in adulthood. The experiments are well designed and results are presented clearly. The authors conclusions are justified by the data. However, some weaknesses that require addressing include: correction of typos; provision of additional methodological details; provision of a more explicit statement as to the novelty of the findings of this work; a more detailed discussion including discussion of the lack of ventral hippocampus-mPFC impairments here versus their previous publication and a discussion on why some behavioural effects (novel object recognition) observed in the juvenile period were not sustained into adulthood.

Comments for the authors:

Given the authors previous publication on the impact of conditional knockdown of AP2γ on hippocampal neurogenesis, anxiety-like behaviour and cognition, the authors need to emphasise what the novelty of this work is and how it advances their previous findings. The translational value and opportunity for therapeutic development is not made explicitly clear and could be commented upon.

Introduction

There are some typos that should be fixed:

L49: "Postnatal mice brain" – please change to mouse brain.

L49 Boldrini paper is on human and not mouse hippocampal neurogenesis and thus should be removed from this sentence.

L53 "in a fined and tuned process" – change to a finely tuned process?

Methods

L318 "distinctive at cohorts"- delete "at"?

Why were only males assessed and not females also?

Open field test – please describe lighting conditions and the size of the area designated as the center of the Open Field.

The FST is normally 6 minutes long with immobility time of the last 4 minutes measured. Why do the authors only measure the last 2 minutes of a 5 minute test? Please provide supporting references for this modification.

Neither the tail suspension test nor the forced swim test are tests of "learned helplessness". Please revise the language used to describe these tests and the behaviour they induce. These tests were developed to measure antidepressant-like behaviour.

Since AP2γ regulates hippocampal neurogenesis , it is curious as to why the authors did not use behavioural tests known to be dependent on hippocampal neurogenesis such as pattern separation or antidepressant action in the novelty suppressed feeding test. This limitation and direction for future research should be acknowledged in the discussion.

Please name/describe the objects used in the novel object recognition test.

Why was morphological assessment of neurons restricted to the dorsal dentate gyrus? I am assuming this is based on the dHi/vHi-mPFC coherence findings but this is not made clear.

Results

Figure 1 – please add the time line and sequence of specific behavioural tests into the figure or describe the sequence and experimental days in which behavioural testing was conducted in the main text.

Fig1G – typo on y-axis – should be DCX and not DXC?

For consistency and to also show the declining rate of neurogenesis during increasing age. Please plot panels F and H, G and I using the same maximum y-axis length i.e. 20 and 15 respectively.

Panel E: Please show representative images from both WT and AP2γ+/- mice.

Please explain the large variation in n numbers across the behavioural tests e.g. in figure 2 and 3.

Fear conditioning – The authors report that AP2γ+/- mice freeze less in context A. This might suggest reduced anxiety which contradicts the OF and EPM findings. This should be discussed.

Figure 5 legend should clearly state whether these measurements were taken from adult or juvenile mice.

There seems to be duplication between Figure 1 and suppl Figure 1 whereby BrdU and BrdU/DCX cell counts and dendritic length are shown twice. Please correct.

Title of suppl Figure 5 should state whether these measurements are made in juvenile or adult mice.

Discussion

The authors should discuss why impairments in NOR in juveniles does not persist to adulthood.

The authors should discuss why in this experiment dHi-mPFC but not vHi-mPFC coherence was disrupted, but in their previous manuscript conditional knockout impaired vHi-mPFC coherence.

L276: "conditional deletion of AP2γ in adulthood lead to a less evident effect on emotional behavior, namely in anxiety-like behavior tested in the OF and EPM behavioral tests, when comparing with the constitutive mice model herein presented (61). This result indicates that constitutive deficiency of AP2γ may exert a longitudinal cumulative impact leading to more severe alterations in behavioral performance of mice". To support this conclusion, the authors should provide empirical data such as percentage difference in time spent in OA of EPM in AP2γ knockdown mice versus controls in both studies or else describe in more specific detail the findings in the previous paper.

[Editors' note: further revisions were suggested prior to acceptance, as described below.]

Thank you for resubmitting your work entitled "Constitutive AP2γ deficiency reduces postnatal hippocampal neurogenesis and induces behavioral deficits in juvenile mice that persist during adulthood" for further consideration by *eLife*. Your revised article has been reviewed by 3 peer reviewers, one of whom is a member of our Board of Reviewing Editors, and the evaluatino has been overseen by Catherine Dulac as the Senior Editor.

The comments below are intended to highlight the results you have achieved. It is to discuss them from a perspective more focused on the ontogeny of behaviors given the systematic comparison that you have undertaken. Moreover, (i) the new "object recognition task" is not a separation task, which does not take away from its interest, on the contrary, and (ii) the proposed formulation for "behavioral despair" is more adapted.

*Reviewer #1:*

In the revised version, the authors addressed most of my comments. However, they have added new behavioral data that deserve to be discussed more deeply in the context of ontogeny of emotion and memory.

1. The authors made several additional experiments. However, some of them have not been included in the revised version. In particular, results of the Figure 2 of the rebuttal letter should be added in figure 3 (in h). These results are really important because they show that animals are too immature to navigate through space using reference memory (Schenk et al. 1985). This is consistent with the delayed development of the hippocampus and indicates that deficits observed at adulthood in the WM are NOT linked to neurons born during the juvenile period or before.

2. The authors demonstrated that constitutive and heterozygous deletion of AP2γ has a complex influence on behaviors:

– Depression-like behaviors remain unchanged

– Some deficits induced in juveniles persist in adulthood : anxiety-like behavior, contextual fear memory (which by the way might be linked to enhanced anxiety levels )

– Some deficits observed in juveniles do not persist in adulthood: novel object recognition

– Some deficits are specific of adult: spatial reference memory, novel object location.

The authors should add value to their results by highlighting the complexity of these observations. This complexity should be discussed in term of ontogeny of behaviors, emergence of the different memory systems and different waves of neurogenesis.

3. There is too much emphasis in the discussion on adult neurogenesis. For example the observation that juveniles exhibit some deficits (anxiety like behavior, simple memory operation) is not relevant to adult neurogenesis! In contrast, the late emergence of spatial reference memory and behavioral flexibility might be! (Dupret et al., 2008; Garthe et al., 2009, see also review of Anacker and Hen R, 2017; Abrous et al., 2021).

4. The title do not reflect the results

5. Abstract/introduction : the concluding sentences do not highlight the novelty of the results.

6. Results

• Ln 109 the age of the animals should be indicated here and not later (ln126).

• Ln 126 The sentence is not clear enough: We assessed the morphology of DCX+ cells *only* in adult animals as during juvenile period (P31), the large amount of DCX+ cells does not allow a proper segmentation of dendrites through this methodology (Figure 2A).

• Ln 150-151: the subtitle does not reflect the results since memory deficits are not systematically observed. Moreover the word cognitive is rather vague (memory is more appropriate).

• The last paragraph of page 6 needs more breath.

• Ln 180: the task used in Figure 3J is not a "behavioral pattern separation" task. In this task, the ability to combine "what" and "where" information is measured (object in context) and mutant mice are impaired in their ability to contextualize the information. Among others, the work of Rasmaran et al., 2016 (Determinants of object-in-context and object-place-context recognition in the developing rat); Barker and Warburton 2011 (When Is the Hippocampus Involved in Recognition Memory). Barker and Warburton 2021 (Putting objects in context: A prefrontal-hippocampal-perirhinal cortex network) might be helpful for the discussion.

• A task measuring "behavioral pattern separation" is not really needed as behavioral flexibility also depends upon adult born neurons (Dupret et al., 2008; Garthe et al., 2009, see also review of Anacker and Hen R, 2017; Abrous et al., 2021).

• Fig3N (and also Fig4L): the authors wrote "No alterations in freezing behavior were observed in context B". These results have not been discussed. The % of freezing are high (40% and 60%) and are certainly higher than that measured before conditioning (please provide the results). Does that mean both groups of animals generalized and did not recognize the context?

• Ln 201 the subtitle does not reflect the results since not all deficits persist (and some appear).

• Ln 222 the 2 contexts are not similar.

• The last paragraph of page 7 needs more breath.

• Ln 218: memory (and not cognitive) performances.

• Ln 227 the authors concluded that "These observations suggest that adult KO mice exhibit deficits in contextual hippocampal", a conclusion that does not fit with the lack of deficits observed in Figure 4H.

• Ln 245 This conclusion does not summarize the results appropriately

7. Methods

• Line 656. Reference memory: please indicate that animals were released for different departure points.

8. Figures

• Figure 2 reading could be improve by aligning:

– Juvenile (P31) hippocampal DG assessment" and "DAPI | DCX"

– Adult ( Age??) hippocampal DG assessment" and "DAPI | DCX"

– DCX+ cells counts and arborization in the adult hippocampal DG : Short dendrites

– DCX+ cells counts and arborization in the adult hippocampal DG : Long dendrites

• Figure 3 G-K : all these results are based on object recognition ("pattern separation "needs to be removed).

• Figure 4 H-J Same comment.

• Figure 3—figure supplement 1: AP2γ constitutive and heterozygous deficiency does not impact sensory motor development.

9. Conclusion

• The discussion focuses too much on adult neurogenesis which is not the critical and new point of this study. The behavioral data and the complex impact of the mutation deserve to be discussed more thoroughly.

*Reviewer #2:*

The authors have responded to all my comments and suggestions. This has resulted, in my opinion, in a manuscript with more clarity. I have one remaining comment:

Behavior during FST and TST

The authors now indicate behavior during the FST and TST as 'behavioral despair'. I disagree with this anthropomorphic interpretation and would like to refer to the following reviews and opinion pieces:

https://pubmed.ncbi.nlm.nih.gov/33955617/

https://pubmed.ncbi.nlm.nih.gov/33548153/

https://pubmed.ncbi.nlm.nih.gov/30738104/

https://pubmed.ncbi.nlm.nih.gov/27034848/

Therefore, I strongly suggest labeling immobile behavior during the FST and TST, in this manuscript and future manuscripts, as 'adaptive behavior to cope with an inescapable stressor'.

*Reviewer #3:*

The authors have addressed all of my previous concerns. The addition of the pattern separation test and novelty-suppressed feeding test has really enhanced the manuscript. The manuscript is very interesting, novel and an important contribution to the field. I very much enjoyed reading it.

---

## [Author Response]

Reviewer #1:[…] It is unclear whether different batches of animals were used for the behavioral testing or alternatively whether the same batch of animals was tested during two different developmental periods. In this latter case, the impact of training during the juvenile period may have interfered with the behavior of animals tested at adulthood.

We thank and totally agree with the reviewer’s suggestion to improve and clarify possible interferences related to a putative repeated behavioral assessment at different timepoints. In this study, individual cohorts of mice were used for behavioral characterization at early postnatal period, juvenile or later at adulthood. Our intention was to avoid any possible impact of habituation or learning from the repetition of behavior assessment, as evoked by the reviewer, and prevent a possible long-term impact of behavioral testing during development periods. In the ”Materials and Methods” section, of the revised manuscript, we added this important information to the reader.

In addition, it would have been interesting to test (1) anxiety before weaning, (2) spatial learning in a WM or contextual fear conditioning in juvenile/adolescent animals.

We acknowledge the reviewer for such pertinent suggestions.

1. We agree that the assessment of the anxiety-like state before weaning represents an additional time point of analysis between the developmental milestones and the behavioral assessment during the juvenile period. As such, in consequence of the reviewer’s suggestion, at postnatal day (P)21, wild-type (WT) and AP2γ KO mice were subjected to the open-field test. Results reveal that similarly to the ones observed during the juvenile period, before weaning, AP2γ KO mice show a significant decrease in distance traveled in the center of the open-field, in comparison to WT animals (Author response image 1). These results reinforce the observation that constitutive and heterozygous deletion of AP2γ promotes an increase of anxiety-like behavior, manifested as early as P21, as shown in Author response image 1. Importantly, AP2γ deletion does not impact the locomotor activity during the juvenile period as denoted by the average velocity in the open-field test (Author response image 1).

**Author response image 1. sa2fig1:** Before weaning, anxiety-like behavior was assessed through the open-field (OF) test. AP2γ KO mice exhibited a decreased in distance traveled in center, when compared to WT , suggesting an anxious-like phenotype (A). No differences in locomotor activity was detected between groups as denoted by a similar average velocity during the test (B). Data presented as mean ± SEM. Sample Size: OF: *n*_WT_ = 10; *n*_AP2γ_ KO = 9. [Student’s t-test].

2. Furthermore, we totally agree with the reviewer’s suggestion to evaluate the impact of constitutive and heterozygous deletion of AP2γ on cognitive performance during the juvenile period, as performed during adulthood. Thus, we performed the Morris water maze (MWM), as also suggested by the editor, and the contextual fear conditioning (CFC) test on WT and AP2γ KO juvenile mice. Results from the CFC reveal that juvenile AP2γ KO mice exhibit a significant decrease in freezing behavior on context probe A when compared to WT animals. No significant alterations in freezing behavior were observed either at context probe B or cue probe. These results are in line with those observed during adulthood, suggesting an early and specific impact of constitutive and heterozygous AP2γ deletion on context-dependent fear memory, observed during the juvenile period and still detected in adult mice. Results of CFC in juvenile mice are now included as figure 3 L-O in the revised manuscript. On the other hand, results from the MWM in juvenile mice were unable to provide further evidence of the impact of AP2γ deletion on cognitive performance, and more particularly on spatial reference memory. Along the days of training, both WT and AP2γ KO mice did not reveal a significant decrease in escape latency to find the hidden platform, suggesting that, at this early age, mice were not able to successfully learn this complex task (Author response image 2). The results of this cognitive task are presented in (Author response image 2) .

**Author response image 2. sa2fig2:** Cognitive performance of juvenile mice in the MWM test. (A) Spatial reference memory was assessed as the average escape latency to find a hidden and fixed platform in each test day. Data presented as mean ± SEM. Sample size: MWM: *n*_WT_ = 10; *n*_AP2γ_KO = 9. [Repeated measures ANOVA].

Additionally, to assess the performance in a cognitive task particularly dependent on hippocampal neurogenesis (Treves et al., 2008), we subjected both experimental groups (WT and AP2γ KO mice) to the pattern separation paradigm during the juvenile period and adulthood. We observed that both juvenile and adult AP2γ KO mice were unable to distinguish changes of objects between distinct contexts as evidenced by a similar exploration of objects in the familiar and novel context. These observations constitute further evidence of a strong impact of the constitutive and heterozygous deletion of AP2γ on multiple cognitive modalities. Results from the pattern separation task are included as figures 3J-K (juvenile) and 4J (adult) in the revised manuscript.

Given that the behavioral syndrome is a stable form from the juvenile period into adulthood, the role of adult hippocampal neurogenesis in this context is unclear.

We thank the reviewer for this comment. In fact, the results of this study reveal that constitutive and heterozygous deletion of AP2γ promotes a significant impact on hippocampal neurogenesis and on behavior which is already evidenced during the juvenile period. In this context, we agree with the reviewer that AP2γ deletion does not impact solely on adult hippocampal neurogenesis but impacts this neurogenic process early on development. As so, throughout the manuscript namely in the “Results” and “Discussion” sections, we clearly state this perspective.

The authors found no differences when assessing DG neuronal characteristics in juvenile and adult mice. It would be interesting to determine whether AP2γ deficiency impacts the dendritic arborisation of Dcx neurons in juvenile and adult mice.

We acknowledge the pertinent reviewer’s suggestion and we agree that assessing the impact of AP2γ deletion on the morphology of immature DCX^+^ neurons would represent a more refine measurement than the morphology of pre-existing mature neurons in the dentate gyrus. In this context, we analyzed the dendritic arborization of DCX^+^ neurons in juvenile and adult mice. We assessed the morphology of short (dendrites extend to the inner molecular layer) and long (dendritic tree reaches the outer molecular layer) DCX^+^ cells. In adult mice, we observed a significant decrease in the arborization of short DCX^+^ cells in AP2γ KO mice when compared to WT. In addition to a decrease in the total dendritic length, we found a decreased complexity of short DCX^+^ cells in AP2γ KO mice denoted by a reduced number of intersections along their dendritic tree. Interestingly, we detected no differences between the dendritic arborization of long DCX^+^ cells in the DG of adult WT and AP2γ KO mice. These observations suggest that in addition to a decrease in the number of DCX^+^ cells (Figure 2C and G), in adult mice, AP2γ deficiency delays the maturation of granular neurons but has no impact on their definitive morphology. In juvenile mice, as expected (Kase et al., 2020; Katsimpardi and Lledo, 2018), we found in both groups a larger amount and density of DCX^+^ cells than at adulthood. In consequence of large overlap of these cells, it was impossible to distinguish individual dendritic trees of DCX^+^ cells which prevent us to analyze their morphology and understand the impact of AP2γ deletion at this period. The analysis of dendritic arborization of DCX^+^ cells in the hippocampal DG of adult mice is now presented in figure 2 of the revised version of the manuscript.

Reviewer #2:[…] I have the following recommendations to improve the clarity of the manuscript.– Adding individual datapoints to the graph would increase clarity for the reader.

We thank the reviewer for this relevant suggestion to improve the clarity of the manuscript. In the revised version of the figures, we now included individual data points to every graph included in this study.

– The rationale behind the current study could be described in greater detail; it could, for example, be better explained why constitutive and heterozygous AP2γ deficiency was studied in this paper.– The reader would also benefit what the information from this paper specifically adds to previous findings form the same group with (mostly) constitutive and homozygous AP2γ deficiency in mice (Mateus-Pinheiro et al. 2017)

We acknowledged the reviewer’s comment and suggestion for a better understanding of the study’s aims. In previous studies, we observed that AP2γ transcription factor was critically required for neuronal specialization and development for specific and defined periods, for proper embryonic development of the cerebral cortex (Pinto et al., 2009), and a continuous generation of the neurons in the adult hippocampal dentate gyrus (Mateus-Pinheiro et al., 2017). In the latter, we used conditional AP2γ knockout mice and showed that conditional deletion of AP2γ in the adult mice also triggered hippocampal-dependent cognitive deficits. In this context, it is important to investigate the cumulative impact of AP2γ deficiency since development. For that we used in this study a constitutive and heterozygous mice model of AP2γ deletion (homozygous mice are not viable) unabling us, for the first time, to assess the importance of AP2γ for postnatal neurodevelopment, particularly on the process of hippocampal neurogenesis, and the impact of its deletion to important hippocampal-dependent behaviors. Furthermore, in light of the importance of AP2γ for the proliferation and expansion of a subpopulation (Tbr2^+^) of transient amplifying progenitors (TAPs), we also intend to reveal the long-term impact of downregulating this cell population since early development. As such, in this study, we applied a longitudinal assessment of postnatal hippocampal neurogenesis and neuronal morphology, regional electrophysiological coherence, and behavioral performance. This extensive and multimodal evaluation of the impact of AP2γ deficiency was performed during the juvenile period and adulthood. Additionally, in the same animal model, we assessed the acquisition of neurodevelopmental milestones early after birth. To understand the cumulative effects of AP2γ deletion, we also intended to determine at which stages of postnatal neurodevelopment AP2γ deficiency was impacting, either at the neurophysiological or/and behavioral level.

Recognizing the importance of frame this study among previous and related publications, in the revised version of the manuscript we made it more clear to the reader.

– The reader would also benefit from detailed information how these mice were exactly generated (e.g. see information on Animals in Mateus-Pinheiro et al. 2017).

We appreciate the suggestion to improve the description of how mice included in this study were generated. In the revised version, in the “Materials and methods” section of the manuscript, it is now described how this mice were generated, obtained and maintained.

– The repeated analysis of coherence or PSD at different frequencies within samples with simple Student t-tests feels incorrect. A repeated measures ANOVA might be more appropriate. At the minimum, a multiple comparison correction should be applied.

We appreciate and agree with the reviewer’s suggestion to change the statistical analysis of the electrophysiological recordings. For the analysis of spectral coherence or power spectral density, we applied, in the revised version of the manuscript, ANOVA Repeated Measures for a more appropriate comparison between groups and the different frequencies.

Reviewer #3:[…] Given the authors previous publication on the impact of conditional knockdown of AP2γ on hippocampal neurogenesis, anxiety-like behaviour and cognition, the authors need to emphasise what the novelty of this work is and how it advances their previous findings. The translational value and opportunity for therapeutic development is not made explicitly clear and could be commented upon.

We thank the reviewer’s comment and suggestion to frame and contextualize this study and enumerate what it adds to previous publications. In previous studies, we observed that AP2γ transcription factor was critically required for neuronal specialization and development for specific and defined periods, for proper embryonic development of the cerebral cortex (Pinto et al., 2009), and a continuous generation of the neurons in the adult hippocampal dentate gyrus (Mateus-Pinheiro et al., 2017). In the latter, conditional deletion of AP2γ in the adult mice also triggered hippocampal-dependent cognitive deficits. In this context, it is important to investigate the cumulative impact of AP2γ deficiency since development. The constitutive and heterozygous mice model of AP2γ deletion (homozygous mice are not viable) unable us to, for the first time, assess the importance of AP2γ for postnatal neurodevelopment, particularly on the process of hippocampal neurogenesis, and the impact of its deletion to important hippocampal-dependent behaviors. Furthermore, in light of the importance of AP2γ for the proliferation and expansion of a subpopulation (Tbr2^+^) of transient amplifying progenitors (TAPs), we also intended to reveal the long-term impact of downregulating this population since early development. As such, in this study, we applied a longitudinal assessment of postnatal hippocampal neurogenesis and neuronal morphology, regional electrophysiological coherence, and behavioral performance. This extensive and multimodal evaluation of the impact of AP2γ deficiency was performed during the juvenile period and at adulthood. Additionally, in the same animal model, we assessed the acquisition of neurodevelopmental milestones early after birth. To understand the cumulative effects of AP2γ deletion, we also intended to determine at which stages of postnatal neurodevelopment AP2γ deficiency was impacting, either at the neurophysiological or/and behavioral level. Thus, through this work, we concluded that constitutive deficiency of AP2γ leads to neurogenic and behavioral impairments since the juvenile period until adulthood, and that these deficits do not occur only in adult mice. With the conditional deletion of AP2γ during adulthood from our study in Molecular Psychiatry 2017 (Mateus-Pinheiro et al., 2017), we were not able to understand how deficiencies of AP2γ transcription factor could modulate postnatal neurodevelopment, particularly hippocampal neurogenesis, and hippocampal-dependent behaviors. Hence, the current study adds a temporal window of analysis not yet studied, and hints that in future works some positive modulation of AP2γ could be applied at younger ages, as a therapeutic approach, to revert deficits in hippocampal neurogenesis in the context of neurological disorders.

Recognizing the importance of frame this study among previous and related publications, in the revised version of the manuscript we made it more clear to the reader.

IntroductionThere are some typos that should be fixed:L49: "Postnatal mice brain" – please change to mouse brain.

As suggested by the reviewer, we increased the accuracy of the first sentence of the introduction section changing the terminology to “mouse brain”.

L49 Boldrini paper is on human and not mouse hippocampal neurogenesis and thus should be removed from this sentence.

We acknowledge the reviewer’s suggestion, and in the revised manuscript we removed the reference Boldrini et., 2018 when referring to the neurogenic process in mice.

L53 "in a fined and tuned process" – change to a finely tuned process?

We appreciate the suggestion and we corrected the sentence in the revised version of the manuscript.

MethodsL318 "distinctive at cohorts" – delete "at"?

Following the reviewer’s suggestion, we deleted ‘at’ from the referred sentence.

Why were only males assessed and not females also?

We thank the reviewer’s comment. As referred to in the subsection ‘animals‘ from the ‘Materials and Materials’, all the animals used in this study were male mice. We acknowledge that it would be interesting to further study the impact of AP2γ constitutive and heterozygous deletion in female mice. In fact, we are currently extending our studies using also female animals. However, taking into consideration the extensive timeline and the multiple cohorts of animals already used for this study, it would not be feasible to double the number of mice. Thus, we decided to restrict it to male mice Moreover, aware of the observed results in emotional-related behaviors, and the impact of estrus cycle in anxiety-like phenotype (Chari et al., 2020; Mora et al., 1996), we would need to either confine the study to female mice with synchronized cycles or add a stressful vaginal smears for cytology.

Open field test – please describe lighting conditions and the size of the area designated as the center of the Open Field.

We appreciate the suggestion to improve the description of the experimental condition of this behavioral test. In the revised version of the manuscript we described the lighting conditions and the size of the defined center of the open-field.

The FST is normally 6 minutes long with immobility time of the last 4 minutes measured. Why do the authors only measure the last 2 minutes of a 5 minute test? Please provide supporting references for this modification.

The reviewer is right. In fact the FST in mice is typically performed in a 6 minutes-long protocol. However, we adapted it to a 5 minutes-long protocol, being the first 3 mins of habituation period and the last 2 mins of test trial as previously described (Mateus-Pinheiro et al., Molecular Psychiatry, 2017). Importantly, the results obtained in this FST protocol are in accordance with those observations from TST, where we found no differences in behavioral despair between WT and AP2γ KO mice.

Neither the tail suspension test nor the forced swim test are tests of "learned helplessness". Please revise the language used to describe these tests and the behaviour they induce. These tests were developed to measure antidepressant-like behaviour.

We thank the reviewer’s comment. In fact, we agree that “learned helplessness” is not the most appropriate term to describe the behavior assessed by tail suspension test and forced swimming test. Thus, throughout the revised manuscript, we changed the term “learned helplessness” to “behavioral despair”.

Since AP2γ regulates hippocampal neurogenesis , it is curious as to why the authors did not use behavioural tests known to be dependent on hippocampal neurogenesis such as pattern separation or antidepressant action in the novelty suppressed feeding test. This limitation and direction for future research should be acknowledged in the discussion.

We appreciate and agree with this pertinent question raised by the reviewer. It is crucial to assess the behavioral impact of the constitutive and heterozygous deletion of AP2γ, a key regulator of adult hippocampal neurogenesis, through behavioral tests particularly dependent on this neurogenic process. As such, we performed the novelty suppressed feeding (NSF) test in both juvenile and adult wild-type and AP2γ KO mice. We observed both in juvenile and adult mice an anxious-like phenotype in AP2γ KO mice, denoted by increased latency to the food pellet in comparison to WT animals. Results from NSF are included in the revised version of the manuscript as figures 3C-D (juvenile) and 4D-E (adult). Furthermore, we assessed the performance of juvenile and adult WT and AP2γ KO mice in the pattern separation task. Results from both periods reveal an impaired capacity of AP2γ KO mice to distinguish changes of objects between distinct contexts as evidenced by a similar exploration of objects in the familiar and novel context. Together, these observations constitute further evidence of the impact of constitutive and heterozygous AP2γ deletion on hippocampal neurogenesis and consequent impairment on dependent behaviors. Results from the pattern separation task are now included as figures 3J-K (juvenile) and 4J (adult) in the revised manuscript.

Please name/describe the objects used in the novel object recognition test.

As suggested by the reviewer, in the revised manuscript we added a detailed description of the objects used in the novel object recognition test.

Why was morphological assessment of neurons restricted to the dorsal dentate gyrus? I am assuming this is based on the dHi/vHi-mPFC coherence findings but this is not made clear.

We acknowledge the reviewer to raise this important point. In our study, the assessment of the neuromorphological impact of constitutive and heterozygous deletion of AP2γ was performed in the dorsal hippocampus. As pointed out by the reviewer, a reason for this option is related to the fact that constitutive and heterozygous deletion of AP2γ solely impacts on the mPFC-dorsal hippocampus coherence. Nevertheless, we cannot rule out a possible impact on the morphology of mature granular neurons in the ventral hippocampus. Indeed, it would be of interest in next studies the assessment of putative changes in number and morphology, particularly of DG neurons in the ventral hippocampus.

ResultsFigure 1 – please add the time line and sequence of specific behavioural tests into the figure or describe the sequence and experimental days in which behavioural testing was conducted in the main text.

We thank the reviewer’s suggestion for clarity on the sequence of behavioral testing used in this study. In the ”Materials and Methods” section of the revised manuscript, we included the order and day of when the behavioral tests were performed, during juvenile and adult periods.

Fig1G – typo on y-axis – should be DCX and not DXC?

As suggested by the reviewer, we corrected this and other typos along the revised version of the manuscript.

For consistency and to also show the declining rate of neurogenesis during increasing age. Please plot panels F and H, G and I using the same maximum y-axis length i.e. 20 and 15 respectively.

We appreciate the reviewer’s suggestion to increase consistency and enable a comparative analysis of neurogenesis in DG with age. As suggested, the quantification of BrdU^+^ and BrdU^+^/DCX^+^ cells for juvenile and adult periods in the revised figures, is now presented with the same y-axis length.

Panel E: Please show representative images from both WT and AP2γ+/- mice.

We followed the suggestion of the reviewer, and we included in panel E of figure 1, images with representative BrdU and DCX stainings in the hippocampal DG of both experimental groups, WT and AP2γ KO mice.

Please explain the large variation in n numbers across the behavioural tests e.g. in figure 2 and 3.

We acknowledge the reviewer for this observation. We agree with the reviewer that there is a variation in the number of animals that performed the different behavioral tests within each period, juvenile or adulthood. This variation results from our intention to prevent overexposure mice with all behavior tests (a total of 13 different tests). We believed this would impact the normal performance of the animals, leading to habituation and training to behavioral testing, and would represent a cofounding effect for the interpretation of the results. Thus, different cohorts of mice containing both experimental groups WT and AP2γ KO were exposed to a representative portion of behavior tests. Each individual cohort was slightly different in their size. Furthermore, some cohorts of mice had commonly performed one or more behavioral tests, which resulted in the combination of data from these cohorts.

Fear conditioning – The authors report that AP2γ+/- mice freeze less in context A. This might suggest reduced anxiety which contradicts the OF and EPM findings. This should be discussed.

We thank the reviewer’s comment. The assessment of cognitive performance in the contextual fear conditioning in both juvenile (now included in the revised version of the manuscript) and adult mice revealed that after one day of training and an acquired light-shock association, AP2γ KO mice display decreased freezing behavior when exposed to the previous context (A), when compared to WT mice. As the context test is performed after the acquisition of context-dependent fear memory, these observations suggest a specific deficit in contextual hippocampal-related memory. In our interpretation, similar observation from exposure to context A that preceded light-shock association would eventually suggest reduced anxiety-like behavior, which would be in line with observations from OF, EPM and NSF.

Figure 5 legend should clearly state whether these measurements were taken from adult or juvenile mice.Title of suppl Figure 5 should state whether these measurements are made in juvenile or adult mice.

We thank the reviewer for these relevant observations. In the legend from the electrophysiological assessment (now Figure 6), we added that these observations were obtained from adult mice. Likewise, the same note was included in the title of Figure 6—figure supplement 1.

There seems to be duplication between Figure 1 and suppl Figure 1 whereby BrdU and BrdU/DCX cell counts and dendritic length are shown twice. Please correct.

We agree that in the previous version of the manuscript, there was a duplication of data in figure 1 and supplementary figure 1, which was removed. In the revised version, the quantification of BrdU^+^ cells and BrdU^+^/DCX^+^ cells in the dentate gyrus either from juvenile and adult mice are included in Figure 1, while neuromorphological analysis of DG granular neurons is included in Figure 2—figure supplement 1.

DiscussionThe authors should discuss why impairments in NOR in juveniles does not persist to adulthood.

We appreciate the reviewer’s suggestion. In fact, juvenile AP2γ KO mice exhibit a decreased performance in the novel object recognition task, when compared to WT mice, which is not observed when assessed at adulthood. Yet, we believe it is worthy to note that adult AP2γ KO mice present a trend towards significance (*p* = 0.08) for cognitive deficits in a similar task when challenged to recognize a change in the location of the object. Furthermore, evidence from other cognitive tests assessed both in juvenile and adult mice, pattern separation and contextual fear conditioning, revealed significant differences between experimental groups at both periods, suggesting that constitutive and heterozygous deletion of AP2γ promotes an early and strong impact on cognitive performance.

The authors should discuss why in this experiment dHi-mPFC but not vHi-mPFC coherence was disrupted, but in their previous manuscript conditional knockout impaired vHi-mPFC coherence.

We thank the reviewer for this pertinent observation. In fact, in this study we sought to understand how constitutive and heterozygous deletion of AP2γ affects the functional connectivity between the hippocampus and the medial prefrontal cortex (mPFC). We observed that the spectral coherence between dorsal hippocampus (dHIP) and mPFC is significantly decreased in delta, theta, and beta frequency bands. This impaired interregional coherence was accompanied by a reduced neuronal activity in the mPFC, including in delta, theta, beta, and low gamma frequencies. We found no differences in the connectivity between the ventral-hippocampus (vHip) and the mPFC. Taking into consideration our previously published work (Mateus-Pinheiro et al., 2017), this was an unexpected result, since conditional deletion of AP2γ in adult mice led to coherence impairments between the vHip-to-mPFC. However, we must take into consideration the time specificity on these mice models of AP2γ deletion. The first and previously published study, deletion of AP2γ was specifically performed at adulthood, while in the current study, mice have a constitutive deficiency of AP2γ since embryonic development. The time in which deficiency of the transcription factor AP2γ is induced might lead to different brain circuitry changes/remodelling, altering for example the connectivity between distinct hippocampal regions and the mPFC, and in this way, contributing to distinct functional readouts as shown in our studies. Although we are in both modulating the hippocampal neurogenic process, we have to bear in mind that AP2γ deficiency since embryonic development might induce not accounted differences contributing to these functional readouts, and the other way around. Yet, it is important to note that similar electrophysiological deficits in dHIP-mPFC coherence were observed in a rat model of hippocampal cytogenesis abrogation, which also denote long-term manifestation of emotional and cognitive deficits (Mateus-Pinheiro et al., 2021).

L276: "conditional deletion of AP2γ in adulthood lead to a less evident effect on emotional behavior, namely in anxiety-like behavior tested in the OF and EPM behavioral tests, when comparing with the constitutive mice model herein presented (Mateus-Pinheiro et al., 2017). This result indicates that constitutive deficiency of AP2γ may exert a longitudinal cumulative impact leading to more severe alterations in behavioral performance of mice". To support this conclusion, the authors should provide empirical data such as percentage difference in time spent in OA of EPM in AP2γ knockdown mice versus controls in both studies or else describe in more specific detail the findings in the previous paper.

We thank the reviewer for this pertinent observation. We agree that the referred statement is not supported by the data of the present study. To fully compare with precision the impact of the conditional and the constitutive heterozygous deletion of AP2γ, a more refine experiment would be needed. In this experiment would be necessary to provide the same experimental conditions for behavioral testing and a similar degree of deletion. In this context, in the revised version of the manuscript, we reformulated this inaccurate statement.

References:

Chari T, Griswold S, Andrews NA, Fagiolini M. 2020. The Stage of the Estrus Cycle Is Critical for Interpretation of Female Mouse Social Interaction Behavior. Front Behav Neurosci 14:1–9. doi:10.3389/fnbeh.2020.00113Kase Y, Kase Y, Shimazaki T, Okano H. 2020. Current understanding of adult neurogenesis in the mammalian brain: How does adult neurogenesis decrease with age? Inflamm Regen. doi:10.1186/s41232-020-00122-xKatsimpardi L, Lledo PM. 2018. Regulation of neurogenesis in the adult and aging brain. Curr Opin Neurobiol. doi:10.1016/j.conb.2018.07.006Mateus-Pinheiro A, Alves ND, Patrício P, Machado-Santos AR, Loureiro-Campos E, Silva JM, Sardinha VM, Reis J, Schorle H, Oliveira JF, Ninkovic J, Sousa N, Pinto L. 2017. AP2γ controls adult hippocampal neurogenesis and modulates cognitive, but not anxiety or depressive-like behavior. Mol Psychiatry 22:1725–1734. doi:10.1038/mp.2016.169Mateus-Pinheiro A, Patrício P, Alves ND, Martins-Macedo J, Caetano I, Silveira-Rosa T, Araújo B, Mateus-Pinheiro M, Silva-Correia J, Sardinha VM, Loureiro-Campos E, Rodrigues AJ, Oliveira JF, Bessa JM, Sousa N, Pinto L. 2021. Hippocampal cytogenesis abrogation impairs inter-regional communication between the hippocampus and prefrontal cortex and promotes the time-dependent manifestation of emotional and cognitive deficits. Mol Psychiatry. doi:10.1038/s41380-021-01287-8Mora S, Dussaubat N, Díaz-Véliz G. 1996. Effects of the estrous cycle and ovarian hormones on behavioral indices of anxiety in female rats. Psychoneuroendocrinology 21:609–620. doi:10.1016/S0306-4530(96)00015-7Pinto L, Drechsel D, Schmid M-T, Ninkovic J, Irmler M, Brill MS, Restani L, Gianfranceschi L, Cerri C, Weber SN, Tarabykin V, Baer K, Guillemot F, Beckers J, Zecevic N, Dehay C, Caleo M, Schorle H, Götz M. 2009. AP2γ regulates basal progenitor fate in a region- and layer-specific manner in the developing cortex. Nat Neurosci 12:1229–1237. doi:10.1038/nn.2399Treves A, Tashiro A, Witter MP, Moser EI. 2008. What is the mammalian dentate gyrus good for? Neuroscience 154:1155–1172. doi:10.1016/j.neuroscience.2008.04.073[Editors' note: further revisions were suggested prior to acceptance, as described below.]

Reviewer #1:In the revised version, the authors addressed most of my comments. However, they have added new behavioral data that deserve to be discussed more deeply in the context of ontogeny of emotion and memory.1. The authors made several additional experiments. However, some of them have not been included in the revised version. In particular, results of the Figure 2 of the rebuttal letter should be added in figure 3 (in h). These results are really important because they show that animals are too immature to navigate through space using reference memory (Schenk et al. 1985). This is consistent with the delayed development of the hippocampus and indicates that deficits observed at adulthood in the WM are NOT linked to neurons born during the juvenile period or before.

Following the reviewer’s suggestion, we included these results in Figure 3 of the revised manuscript.

2. The authors demonstrated that constitutive and heterozygous deletion of AP2γ has a complex influence on behaviors:– Depression-like behaviors remain unchanged– Some deficits induced in juveniles persist in adulthood : anxiety-like behavior, contextual fear memory (which by the way might be linked to enhanced anxiety levels )– Some deficits observed in juveniles do not persist in adulthood: novel object recognition– Some deficits are specific of adult: spatial reference memory, novel object location.The authors should add value to their results by highlighting the complexity of these observations. This complexity should be discussed in term of ontogeny of behaviors, emergence of the different memory systems and different waves of neurogenesis.

We thank the reviewer for raising such a pertinent point. Indeed, constitutive and heterozygous deletion of AP2γ transcription factor leads to complex behavior alterations in juvenile and adult animals. We acknowledge this complexity in the discussion of the revised version of the manuscript.

3. There is too much emphasis in the discussion on adult neurogenesis. For example the observation that juveniles exhibit some deficits (anxiety like behavior, simple memory operation) is not relevant to adult neurogenesis! In contrast, the late emergence of spatial reference memory and behavioral flexibility might be! (Dupret et al., 2008; Garthe et al., 2009, see also review of Anacker and Hen R, 2017; Abrous et al., 2021).

We believe these two pertinent points (2 and 3) raised by the reviewer are interconnected. As so, we decided to address the arguments together.

In the last years, we have seen an increased number of studies focusing on the functional specificity of hippocampal neurogenesis during development (DHN) and adult hippocampal neurogenesis (AHN). The ontogenetic interpretation of hippocampal neurogenesis assigns different functional relevance to DHN (as the neurogenic process that establishes the basic repertoire of adaptable behaviors) and AHN (as the neurogenic process that underpins the adult brain's ability to adapt functional behaviors) (Abrous et al., 2021). This functional dissociation between these two types of hippocampal neurogenesis is very interesting because it allows us to understand how influences in the neurogenic process at different stages, whether during development or adulthood, can lead to distinct impairments in emotional states and cognitive functions.

AP2γ transcription factor plays detrimental roles in early mammalian extraembryonic development and organogenesis, being one of the molecular components regulating the number of upper layer neurons during ontogeny and phylogeny in the developing cortex (Pinto et al., 2009). Also, AP2γ expression persists in the adult hippocampus, acting as a promoter of proliferation and neuronal differentiation (Mateus-Pinheiro et al., 2017). In this context, we sought to comprehend how defects in the neurogenic process since the early development through constitutive and heterozygous deletion of AP2γ impact function and behavior in specific developmental stages: early postnatal, juvenile phase, and adulthood. Our findings revealed that constitutive and heterozygous AP2γ deletion has a significant impact on hippocampal neurogenesis and behavioral dimensions in juvenile and adult mice, without affecting sensory-motor development. We agree with the reviewer that AP2γ deletion impacts neurogenesis earlier in development rather than just adult hippocampal neurogenesis. As so, throughout the manuscript, namely in the Results and Discussion sections, we clearly state this perspective.

Memory and emotional responses are not distinct ontogenetic processes, but they progress from simple to complex capacities (Abrous et al., 2021). We can hypothesize that AP2γ transcription factor deficiencies can alter the ontogeny of the hippocampal neurogenic process, altering the development of behavior complexity associated with the hippocampus. To address this possibility, we would need to trace the early postnatally born hippocampal neurons, as Marie Lods and colleagues previously performed (Lods et al., 2021), and infer their activity throughout the juvenile phase and adulthood. Moreover, we would need to discriminate how AP2γ transcription factor deficiencies distinctly impair DHN and AHN and their functional outputs. In this study, we observed that at both the juvenile phase and during adulthood hippocampal neurogenesis is reduced, being the transcriptional network underlying this neurogenic process impaired. Furthermore, we noticed that adult AP2γ KO mice showed not only reduced hippocampal neurogenesis but also presented delayed maturation of granular neurons. However, we did not follow the activation/inactivation of these granular neurons throughout different behavioral tests, nor did we analyze how AP2γ transcription factor deficiencies could impact dentate gyrus engrams differently during the juvenile phase and adulthood.

Nonetheless, as AP2γ KO mice have decreased hippocampal neurogenesis since early development, the basic repertoire of adaptive behaviors might be already compromised, leading to behavioral deficits as early as the juvenile phase. Impairments in early postnatal neurogenesis induced by deficiencies in AP2γ transcription factor contributes to a hypersensitivity towards aversive stimulus, as uncovered by the presence of an anxious-like phenotype, and memory impairments as revealed by the AP2γ KO mice performance in multiple cognitive tests. Moreover, the adult emergence of behavioral flexibility by AHN (Abrous et al., 2021) and the impaired performance of adult AP2γ KO mice in anxious behavior testing, contextual memory, and cognitive behavior flexibility demonstrate the importance of an intact adult hippocampal neurogenic process for such functional outputs.

Recognizing the reviewer's comments and the fact that some concepts had not yet been fully explored in our work, we further detailed them in the revised manuscript.

4. The title do not reflect the results

We understand the reviewer's concerns about the title, and we revised it to reflect the findings of our study.

5. Abstract/introduction : the concluding sentences do not highlight the novelty of the results.

As implied by the reviewer, we updated both the concluding sentences of the Abstract and Introduction.

6. Results• Ln 109 the age of the animals should be indicated here and not later (ln126).

Following the reviewer’s suggestion, we added the animals’ age at this point (ln125).

• Ln 126 The sentence is not clear enough: We assessed the morphology of DCX+ cells only in adult animals as during juvenile period (P31), the large amount of DCX+ cells does not allow a proper segmentation of dendrites through this methodology (Figure 2A).

We appreciate the advice to make the results presentation clearer. We added the word "only" in response to the reviewer's suggestion (ln142).

• Ln 150-151: the subtitle does not reflect the results since memory deficits are not systematically observed. Moreover the word cognitive is rather vague (memory is more appropriate).

We understand the reviewer's concern about the subtitle, and we revised it to reflect the findings specifically to these results.

• The last paragraph of page 6 needs more breath.

Following the reviewer’s suggestion, in the revised manuscript, we updated this paragraph.

• Ln 180: the task used in Figure 3J is not a "behavioral pattern separation" task. In this task, the ability to combine "what" and "where" information is measured (object in context) and mutant mice are impaired in their ability to contextualize the information. Among others, the work of Rasmaran et al., 2016 (Determinants of object-in-context and object-place-context recognition in the developing rat); Barker and Warburton 2011 (When Is the Hippocampus Involved in Recognition Memory). Barker and Warburton 2021 (Putting objects in context: A prefrontal-hippocampal-perirhinal cortex network) might be helpful for the discussion.• A task measuring "behavioral pattern separation" is not really needed as behavioral flexibility also depends upon adult born neurons (Dupret et al., 2008; Garthe et al., 2009, see also review of Anacker and Hen R, 2017; Abrous et al., 2021).

We thank the reviewer for such pertinent correction. In the revised manuscript, we updated this behavioral paradigm nomenclature and reference. Moreover, we further discussed the relevance of these behavioral findings in the context of our work and for the field.

• Fig3N (and also Fig4L): the authors wrote "No alterations in freezing behavior were observed in context B". These results have not been discussed. The % of freezing are high (40% and 60%) and are certainly higher than that measured before conditioning (please provide the results). Does that mean both groups of animals generalized and did not recognize the context?

To address the reviewer's concern, we included the results of the freezing behavior analysis before conditioning (Figure 3N and Figure 4K). As expected, freezing behavior prior to conditioning was low, both during the juvenile phase and adulthood. WT and AP2γ KO mice identically explored the acrylic cylinder in which they were placed, denoted by the low levels of freezing behavior.

Concerning the freezing behavior in Context B, the percentages were lower than those observed in the conditioning context and light-cued probes. However, as referred by the reviewer, the generalization and the misplaced recognition of the new context after the conditioning could be a concerning observation. Nonetheless, the difference in the percentage of freezing of WT mice between Context A and B is of relevancy. WT mice demonstrated a lower average percentage of freezing in Context B during the juvenile and adulthood evaluations. These findings show that WT mice do not generalize and instead recognize Context B as new. Both in the juvenile and adult phases, the freezing percentages in context B were still lower than the ones observed in the conditioning contexts and light-cued probes.

• Ln 201 the subtitle does not reflect the results since not all deficits persist (and some appear).

We understand the reviewer’s concern about the subtitle, and we revised it to reflect the observed findings.

• Ln 222 the 2 contexts are not similar.

The reviewer is correct. In fact, the two contexts are not similar. The different contexts were identical solely in terms of behavioral paradigm (object disposition and assessment). In the revised manuscript, we corrected this suggestion.

• The last paragraph of page 7 needs more breath.

Following the reviewer’s suggestion, in the revised manuscript, we updated this paragraph.

• Ln 218: memory (and not cognitive) performances.

We updated this term in the revised manuscript in response to the reviewer's suggestion.

• Ln 227 the authors concluded that "These observations suggest that adult KO mice exhibit deficits in contextual hippocampal", a conclusion that does not fit with the lack of deficits observed in Figure 4H.

In fact, despite a trend towards a decreased preference to explore the displaced object (Figure 4H), adult AP2γ KO animals present no differences in the preference for the novel object in the object recognition test (ORT) (Figure 4I) when compared to the WT group. This was quite surprising since, at the juvenile window of analysis, this was already an established phenotype.

Memory and emotional responses are not singular ontogenetic processes, but they progress from simple to complex capacities (Abrous et al., 2021). During rodents’ adulthood, it is already well described that animals in normal/healthy conditions can recognize novel objects, locations and discriminate objects in different contexts (Abrous et al., 2021; Ainge and Langston, 2012). Moreover, these abilities emerge early in the juvenile phase (around P17) and reach maturity over adolescence (Ramsaran et al., 2016).

In the behavior assessment at juvenile period performed in this work, we confirmed that the WT mice were able to recognize the novel object and its displacement. In these ORT tasks, WT animals displayed an average exploration of the displaced and novel object as high as 80%, with AP2γ KO mice showing impaired performance in the novel object recognition. During adulthood, in the novel location task, WT mice showed similar percentages of exploration of the displaced object (~80%). However, in the recognition task, the exploration of the novel object dropped under 60%. Nevertheless, results from the Object-in-context (OIC) tasks and the Contextual Fear Conditioning (CFC) test point out that AP2γ constitutive deletion induced impairments in information contextualization. In OIC, both during the juvenile phase and adulthood, WT mice spent a greater proportion of time exploring the out of context object than the object in the familiar environment, whereas AP2γ KO mice showed no preference for object exploration (Figure 3K and Figure 4J). Moreover, in the CFC behavioral paradigm, juvenile and adult AP2γ KO mice exhibited reduced freezing behavior when exposed to the conditioning context where they have previously received foot shocks (Figure 3O and Figure 4L). These results reinforce the idea that deletion of the AP2γ transcription factor induces contextual memory impairments.

We understand the reviewers’ point of view, regarding the lack of deficits in this specific novel object exploration task. However, we believe that with the results obtained in the previous round of revisions in the OIC paradigm, as well as with the addition of the CFC evaluation at the juvenile phase, we were able to corroborate our conclusion. Taking into consideration this pertinent point raised by the reviewer, we will further detail our perspective in the Results section of the manuscript (ln215-217 and ln251-253).

• Ln 245 This conclusion does not summarize the results appropriately

Following the reviewer’s suggestion, in the revised manuscript, we changed the referred paragraph.

7. Methods• Line 656. Reference memory: please indicate that animals were released for different departure points.

As suggested by the reviewer, we added the information about the release departure points to the reference memory task description.

8. Figures• Figure 2 reading could be improve by aligning:– Juvenile (P31) hippocampal DG assessment" and "DAPI | DCX"– Adult ( Age??) hippocampal DG assessment" and "DAPI | DCX"– DCX+ cells counts and arborization in the adult hippocampal DG : Short dendrites– DCX+ cells counts and arborization in the adult hippocampal DG : Long dendrites• Figure 3 G-K : all these results are based on object recognition ("pattern separation "needs to be removed).• Figure 4 H-J Same comment.• Figure 3—figure supplement 1: AP2 constitutive and heterozygous deficiency does not impact sensory motor development.

We thank the reviewer for these suggestions to improve the clarity of the Main and Supplementary Figures. Thus, in the revised version of the manuscript, we followed the indications herein presented.

9. Conclusion• The discussion focuses too much on adult neurogenesis which is not the critical and new point of this study. The behavioral data and the complex impact of the mutation deserve to be discussed more thoroughly.

We thank the reviewer for the suggestions given throughout the reviewing process. In the revised manuscript, we updated the Discussion and Conclusion considering all indications herein presented.

Reviewer #2:The authors have responded to all my comments and suggestions. This has resulted, in my opinion, in a manuscript with more clarity. I have one remaining comment:Behavior during FST and TSTThe authors now indicate behavior during the FST and TST as 'behavioral despair'. I disagree with this anthropomorphic interpretation and would like to refer to the following reviews and opinion pieces:https://pubmed.ncbi.nlm.nih.gov/33955617/https://pubmed.ncbi.nlm.nih.gov/33548153/https://pubmed.ncbi.nlm.nih.gov/30738104/https://pubmed.ncbi.nlm.nih.gov/27034848/Therefore, I strongly suggest labeling immobile behavior during the FST and TST, in this manuscript and future manuscripts, as 'adaptive behavior to cope with an inescapable stressor'.

We acknowledge the reviewer for such pertinent suggestions, as well as the references shared with us. We understand this point of view regarding FST and TST and how the analysis of these paradigms has been shifting in the last years (Gorman‐Sandler and Hollis, 2021; Molendijk and de Kloet, 2019; Molendijk and Kloet, 2021). Thus, throughout the revised manuscript, we changed the term “behavioral despair” to either “adaptive behavior to cope with an inescapable stressor” or “coping behavior”. In some contexts, we had to summarize the suggested terminology for a simpler version. Nevertheless, we always referred that this coping mechanism is related to an inescapable stressor (either the water in the FST or the suspension in the TST).

References:

Abrous DN, Koehl M, Lemoine M. 2021. A Baldwin interpretation of adult hippocampal neurogenesis: from functional relevance to physiopathology. Mol Psychiatry 1–20. doi:10.1038/s41380-021-01172-4

Ainge JA, Langston RF. 2012. Ontogeny of neural circuits underlying spatial memory in the rat. Front Neural Circuits 6:1–10. doi:10.3389/fncir.2012.00008

Gorman‐Sandler E, Hollis F. 2021. The forced swim test: Giving up on behavioral despair. Eur J Neurosci ejn.15270. doi:10.1111/ejn.15270

Lods M, Pacary E, Mazier W, Farrugia F, Mortessagne P, Masachs N, Charrier V, Massa F, Cota D, Ferreira G, Abrous DN, Tronel S. 2021. Adult-born neurons immature during learning are necessary for remote memory reconsolidation in rats. Nat Commun 12:1778. doi:10.1038/s41467-021-22069-4

Mateus-Pinheiro A, Alves ND, Patrício P, Machado-Santos AR, Loureiro-Campos E, Silva JM, Sardinha VM, Reis J, Schorle H, Oliveira JF, Ninkovic J, Sousa N, Pinto L. 2017. AP2γ controls adult hippocampal neurogenesis and modulates cognitive, but not anxiety or depressive-like behavior. Mol Psychiatry 22:1725–1734. doi:10.1038/mp.2016.169

Molendijk ML, de Kloet ER. 2019. Coping with the forced swim stressor: Current state-of-the-art. Behav Brain Res 364:1–10. doi:10.1016/j.bbr.2019.02.005

Molendijk ML, Kloet ER. 2021. Forced swim stressor: Trends in usage and mechanistic consideration. Eur J Neurosci ejn.15139. doi:10.1111/ejn.15139

Pinto L, Drechsel D, Schmid M-T, Ninkovic J, Irmler M, Brill MS, Restani L, Gianfranceschi L, Cerri C, Weber SN, Tarabykin V, Baer K, Guillemot F, Beckers J, Zecevic N, Dehay C, Caleo M, Schorle H, Götz M. 2009. AP2γ regulates basal progenitor fate in a region- and layer-specific manner in the developing cortex. Nat Neurosci 12:1229–1237. doi:10.1038/nn.2399

Ramsaran AI, Westbrook SR, Stanton ME. 2016. Ontogeny of object-in-context recognition in the rat. Behav Brain Res 298:37–47. doi:10.1016/j.bbr.2015.04.011